# D²O: A Dual Debiasing Operator for Training-Free Test-Time Adaptation of Vision–Language Models

Yihong Luo [1]  Wenwu He [1 2]  Dong Liang [3 4]  Yihang Zhou [3]  Zhuo-Xu Cui [3 4]

## Abstract

Training-free test-time adaptation (TTA) for vision-language models (VLMs) can improve zero-shot classification under mild shifts, but often degrades under severe style/environment variation. We identify two shared failure modes in training-free pipelines: (i) retrieval confounding, where feature similarity is dominated by nuisance/style variation and corrupts retrieval evidence; and (ii) environment-biased priors, where VLM logits exhibit environment-dependent centered shifts that distort gating and prior-like terms. Therefore, we propose D²O, a training-free debiasing operator that outputs three inference-time objects: a retrieval-oriented content feature for semantic matching, a style-aware routing coordinate for bias tracking, and debiased logits for corrected priors. D²O composes plug-and-play with retrieval-based and closed-form Gaussian adapters in online and transductive settings. We further provide operator-to-decision guarantees: finite-difference covariance recovers a nuisance-sensitive subspace, routing-based EMA controls centered-logit bias estimates, and these errors yield bounded posterior log-odds perturbations, leading to a margin-based condition for label invariance. Extensive experiments show that D²O achieves its clearest gains under style/environment-dominant shifts, with broader gains elsewhere. Code is available at https://github.com/MAiTL-Group/D2O.

## 1. Introduction

Test-time adaptation (TTA) aims to improve robustness when a deployed model encounters distribution shifts at test time. In this work, we study *training-free* TTA for vision-language models (VLMs) (Jia et al., 2021; Yuan et al., 2021; Fang et al., 2023; Zhang et al., 2024b), where adaptation must be performed without source data, backpropagation, or parameter updates. We consider two inference-time settings: an **online** setting, where unlabeled test instances are processed sequentially after a short initialization/calibration buffer from the test stream, and a **transductive** setting, where the entire unlabeled test set is available offline and predictions are produced by inference-time computation.

Although recent training-free adapters—including cache-based retrieval methods and closed-form Gaussian adapters—can improve robustness under mild shifts, their performance often degrades sharply under strong style or environment variations (Koh et al., 2021; Guan & Liu, 2022; Liang et al., 2025). We attribute this failure to two structural confounders shared across training-free pipelines. First, **retrieval confounding** arises because similarity in the VLM feature space can be dominated by nuisance or style cues, causing retrieved neighbors to be environment-matched rather than class-matched and thereby corrupting retrieval evidence. Second, **environment-biased priors** occur when VLM logits exhibit environment-specific centered shifts, which can mislead entropy gating, pseudo-labeling, and prior-like terms commonly used by training-free adapters. Figure 1 illustrates these failure modes by contrasting representative CLIP-based adaptation paradigms.

Motivated by this diagnosis, we take an operator-level viewpoint: instead of redesigning each adapter family, we debias the inference-time objects used by training-free adapters. Rather than pursuing perfect style–content disentanglement, we use an inference-oriented decomposition: style-aware signals support environment routing and bias tracking, retrieval-oriented content signals reduce style-confounded retrieval, and raw features are retained where their distributional statistics remain useful.

Concretely, we introduce D²O, a plug-and-play *debiasing*

[1]School of Computing and Data Science, Fujian University of Technology, Fuzhou, Fujian, China [2]Fujian Provincial Key Laboratory of Big Data Mining and Applications, Fuzhou, Fujian, China [3]Shenzhen Institutes of Advanced Technology, Chinese Academy of Sciences, Shenzhen, Guangdong, China [4]Guangdong Provincial Key Laboratory of Multimodality Non-Invasive Brain-Computer Interfaces, Shenzhen, Guangdong, China. Correspondence to: Wenwu He <hwwhbb@163.com>, Yihang Zhou <yh.zhou2@siat.ac.cn>, Zhuo-Xu Cui <zx.cui@siat.ac.cn>.

*Proceedings of the 43rd International Conference on Machine Learning*, Seoul, South Korea. PMLR 306, 2026. Copyright 2026 by the author(s).

**Figure 1.** **Overview of three CLIP-based adaptation paradigms.** (a) Test-time prompt tuning updates learnable prompts via backpropagation using CLIP predictions on augmented test views. (b) Cache-based training-free TTA (e.g., TDA/ADAPT) maintains a key-value cache and fuses retrieval-based outputs with the CLIP prediction. (c) D²O outputs three inference-time objects—a retrieval-oriented content feature, a style-aware routing coordinate, and debiased logits—uses the retrieval-oriented content feature to drive cache retrieval, and applies routing-based centered-logit bias correction to the CLIP prior before producing the adapted prediction.

*operator* (Figure 1c) that maps each test sample $\mathbf{x}$ to

$$\big(\mathbf{f}_{\mathrm{cnt}}(\mathbf{x}),\ \mathbf{z}_{\mathrm{sty}}(\mathbf{x}),\ \mathbf{s}_{\mathrm{deb}}(\mathbf{x})\big),$$

where $\mathbf{f}_{\mathrm{cnt}}(\mathbf{x})$ is a retrieval-oriented content feature that suppresses nuisance-sensitive directions for semantic matching, $\mathbf{z}_{\mathrm{sty}}(\mathbf{x})$ is a style-aware routing coordinate for environment-aware bias tracking, and $\mathbf{s}_{\mathrm{deb}}(\mathbf{x})$ is a debiased logit vector that provides a corrected prior. D²O is strictly training-free and can be inserted into existing inference pipelines without changing their internal learning mechanics.

We instantiate D²O on two representative host families. **D²O+TDA** improves retrieval-based adaptation by using $\mathbf{f}_{\mathrm{cnt}}(\mathbf{x})$ as the retrieval key and $\mathbf{s}_{\mathrm{deb}}(\mathbf{x})$ as a corrected prior. **D²O+ADAPT** combines (i) a debiased prior from $\mathbf{s}_{\mathrm{deb}}(\mathbf{x})$, (ii) a closed-form Gaussian posterior computed from a bank, and (iii) a retrieval-oriented bank evidence term computed via $\mathbf{f}_{\mathrm{cnt}}(\mathbf{x})$. The same bank mechanism supports both online updates and transductive global selection. Empirically, D²O is most beneficial under style/environment-dominant shifts, while providing smaller but consistent gains on standard OOD, corruption, and fine-grained benchmarks.

Our contributions are summarized as follows:

1. **D²O operator.** We propose a strictly training-free, plug-and-play debiasing operator that produces a retrieval-oriented content feature $\mathbf{f}_{\mathrm{cnt}}(\mathbf{x})$, a style-aware routing coordinate $\mathbf{z}_{\mathrm{sty}}(\mathbf{x})$, and debiased logits $\mathbf{s}_{\mathrm{deb}}(\mathbf{x})$ at inference time to mitigate retrieval confounding and environment-biased priors.

2. **Cross-host instantiations in online and transductive settings.** We instantiate D²O on two representative training-free host families—retrieval-based adaptation and Gaussian posterior adaptation—and show that both can operate in online and transductive TTA through a unified bank protocol.

3. **Operator-to-decision theory.** We provide identifiability and robustness guarantees that propagate subspace recovery and bias estimation errors to decision-relevant

log-odds perturbations, yielding margin-based sufficient conditions for label invariance.

## 2. Method

We present D²O as a training-free *debiasing operator* for inference under style/environment shift. Rather than re-designing a specific host adapter, D²O acts on the inference-time objects that are corrupted differently under distribution shift. Given a test sample $\mathbf{x}$, it produces three sample-specific quantities: a style-aware routing coordinate $\mathbf{z}_{\mathrm{sty}}(\mathbf{x})$, a retrieval-oriented content feature $\mathbf{f}_{\mathrm{cnt}}(\mathbf{x})$, and a debiased logit vector $\mathbf{s}_{\mathrm{deb}}(\mathbf{x})$. The operator consists of four steps: (1) estimate a nuisance-sensitive subspace from a short warm-up buffer, (2) construct an inference-oriented decomposition into style-aware routing and retrieval-oriented content signals, (3) track cluster-wise centered-logit bias in the routing space, and (4) return debiased inference objects to the downstream host adapter. Importantly, D²O does *not* aim at perfect style–content disentanglement; instead, it seeks an inference-oriented style–content decomposition that is useful for test-time debiasing, as summarized in Algorithm 1.

### 2.1. Preliminaries

**VLM backbone.** Contrastive Language–Image Pretraining (CLIP) (Radford et al., 2021) is a vision-language model trained with contrastive learning on a large-scale corpus of image–text pairs. It maps images and texts into a shared embedding space and aligns their representations via a contrastive objective, thereby enabling strong zero-shot transfer across a broad range of vision-language tasks. We let $E_{\mathrm{img}}(\cdot)$ and $E_{\mathrm{txt}}(\cdot)$ denote the image and text encoders, and let $\mathcal{N}(\cdot)$ denote $\ell_2$ normalization. Given a test sample $\mathbf{x}$, we compute its normalized image feature as

$$\mathbf{f}(\mathbf{x}) \leftarrow \mathcal{N}(E_{\mathrm{img}}(\mathbf{x})) \in \mathbb{R}^d. \tag{1}$$

Let $\{\mathbf{t}_k\}_{k=1}^{C}$ be normalized text prototypes constructed from prompts, where $C$ denotes the number of classes. The stan-

dard zero-shot CLIP logits are

$$\mathbf{s}_{\mathrm{clip}}(\mathbf{x})_k = \frac{1}{\tau} \mathbf{t}_k^\top \mathbf{f}(\mathbf{x}), \qquad k \in \{1, \dots, C\}, \qquad (2)$$

where $\tau > 0$ is the temperature.

## 2.2. Warm-up: nuisance-sensitive subspace from finite-difference responses

We first draw a warm-up set $\{\mathbf{x}_i\}_{i=1}^{N_S}$. In the online setting, this buffer is taken from the same evaluation stream and is used only to initialize the operator before sequential adaptation. Let $\mathcal{T}$ be a weak augmentation family with $A$ paired perturbation types $\{T_{+\epsilon}^{(a)}, T_{-\epsilon}^{(a)}\}_{a=1}^A$ for a small $\epsilon > 0$.

The key idea is that these paired weak perturbations largely preserve class identity while inducing nuisance and style variations. Therefore, their finite-difference responses emphasize *nuisance-sensitive directions* rather than semantic identity. For each pair, we compute

$$\boldsymbol{\delta}_{i,a} = \frac{\mathcal{N}\Big(E_{\mathrm{img}}(T_{+\epsilon}^{(a)} \mathbf{x}_i)\Big) - \mathcal{N}\Big(E_{\mathrm{img}}(T_{-\epsilon}^{(a)} \mathbf{x}_i)\Big)}{2\epsilon} \in \mathbb{R}^d. \tag{3}$$

We stack all responses into $\mathbf{D} \in \mathbb{R}^{N_\delta \times d}$, where $N_\delta = N_S A$, and form the response covariance

$$\mathbf{C} = \frac{1}{N_\delta} \mathbf{D}^\top \mathbf{D}. \tag{4}$$

Let $\widehat{\mathbf{V}} \in \mathbb{R}^{d \times r}$ be the top-$r$ eigenvectors of $\mathbf{C}$, and define the associated projector

$$\widehat{\mathbf{P}} = \widehat{\mathbf{V}} \widehat{\mathbf{V}}^\top. \tag{5}$$

The resulting subspace is *not* intended to provide a complete representation of style. Instead, it serves as a low-dimensional nuisance-sensitive subspace for style-aware routing and bias tracking under shift. The finite-difference augmentation types and default parameter settings are summarized in Appendix E.1.

## 2.3. Per-sample retrieval–routing decomposition

For a test sample $\mathbf{x}$, we use the feature $\mathbf{f}(\mathbf{x})$ from Eq. (1). Given the estimated basis $\widehat{\mathbf{V}}$, we project $\mathbf{f}(\mathbf{x})$ onto the nuisance-sensitive subspace and use the resulting coordinates as a style-aware routing coordinate:

$$\mathbf{z}_{\mathrm{sty}}(\mathbf{x}) = \widehat{\mathbf{V}}^\top \mathbf{f}(\mathbf{x}) \in \mathbb{R}^r, \tag{6}$$

Here, $\mathbf{z}_{\mathrm{sty}}(\mathbf{x})$ should be interpreted as a *style-aware routing coordinate* for bias tracking during adaptation, rather than a complete style representation.

To obtain a *retrieval-oriented content feature*, we softly suppress the projected nuisance component from $\mathbf{f}(\mathbf{x})$ and then renormalize:

$$\mathbf{f}_{\mathrm{cnt}}(\mathbf{x}) = \mathcal{N}\Big(\mathbf{f}(\mathbf{x}) - \lambda_{\mathrm{proj}}\widehat{\mathbf{P}}\,\mathbf{f}(\mathbf{x})\Big), \qquad \lambda_{\mathrm{proj}} \in (0, 1]. \tag{7}$$

Thus, $\mathbf{f}_{\mathrm{cnt}}(\mathbf{x})$ is not a perfectly style-free or fully disentangled representation; rather, it is a retrieval-oriented content feature that is more suitable for semantic retrieval because nuisance-sensitive directions are suppressed. This suppression reduces style-dominated retrieval while largely preserving semantic information.

## 2.4. Environment routing and centered-logit debiasing

To track non-stationary test environments, we maintain $K_{\mathrm{env}}$ online clusters in the routing space, each storing a centroid $\mathbf{c}_k \in \mathbb{R}^r$ and a bias template $\mathbf{b}^{(k)} \in \mathbb{R}^C$. Given the style-aware routing coordinate $\mathbf{z}_{\mathrm{sty}}(\mathbf{x})$, we assign each sample to its nearest centroid:

$$k^*(\mathbf{x}) = \underset{k \in \{1, \dots, K_{\mathrm{env}}\}}{\arg\min} \|\mathbf{z}_{\mathrm{sty}}(\mathbf{x}) - \mathbf{c}_k\|_2^2. \tag{8}$$

**Centroid update.** After routing, we update the assigned centroid using an online running average. Let $n_{k^*(\mathbf{x})}$ be the current number of samples assigned to cluster $k^*(\mathbf{x})$. Then

$$\mathbf{c}_{k^*(\mathbf{x})} \leftarrow \frac{n_{k^*(\mathbf{x})}\, \mathbf{c}_{k^*(\mathbf{x})} + \mathbf{z}_{\mathrm{sty}}(\mathbf{x})}{n_{k^*(\mathbf{x})} + 1}. \tag{9}$$

**Centered logits.** We are interested in environment-dependent *relative class distortion*, rather than a uniform confidence shift shared by all classes. Accordingly, we center the logits by removing their mean component. For any $\mathbf{s} \in \mathbb{R}^C$, we define

$$\mathcal{C}(\mathbf{s}) = \mathbf{s} - \frac{1}{C}(\mathbf{1}^\top \mathbf{s})\mathbf{1}, \tag{10}$$

Centering removes the sample-level confidence offset while retaining the class-relative distortion that can bias pseudo-labeling and prior-like terms. Hence, $\mathcal{C}(\mathbf{s}_{\mathrm{clip}}(\mathbf{x}))$ should be interpreted as the relative class-preference pattern in centered-logit space.

**EMA update in centered-logit space.** We update the bias template of the routed cluster with an exponential moving average (EMA):

$$\mathbf{b}^{(k^*(\mathbf{x}))} \leftarrow (1-\rho)\mathbf{b}^{(k^*(\mathbf{x}))} + \rho\, \mathcal{C}(\mathbf{s}_{\mathrm{clip}}(\mathbf{x})), \qquad \rho \in (0, 1]. \tag{11}$$

The EMA template accumulates the average centered-logit pattern of samples routed to the same environment cluster, thereby estimating an environment-specific bias. Thus, $\mathbf{b}^{(k)}$ is maintained in centered-logit rather than raw-logit space.

**Debiased logits (corrected prior).** Finally, we subtract the estimated cluster-wise bias from the original zero-shot logits to obtain a corrected prior:

$$\mathbf{s}_{\text{deb}}(\mathbf{x}) = \mathbf{s}_{\text{clip}}(\mathbf{x}) - \gamma_{\text{env}}\, \mathbf{b}^{(k^*(\mathbf{x}))}, \qquad \gamma_{\text{env}} \geq 0. \quad (12)$$

---

**Algorithm 1** D²O (training-free debiasing operator)

---

1: **Input:** image encoder $E_{\text{img}}$ and zero-shot logits $\mathbf{s}_{\text{clip}}(\mathbf{x})$; weak paired augmentations $\mathcal{T}$; warm-up size $N_S$; style–nuisance rank $r$; clusters $\{\mathbf{c}_k\}_{k=1}^{K_{\text{env}}}$ and bias templates $\{\mathbf{b}^{(k)}\}_{k=1}^{K_{\text{env}}}$ (initialized to $\mathbf{0}$); hyperparameters $\epsilon, \rho, \gamma_{\text{env}}, \lambda_{\text{proj}}$

2: **Output:** retrieval-oriented content feature $\mathbf{f}_{\text{cnt}}(\mathbf{x})$, style-aware routing coordinate $\mathbf{z}_{\text{sty}}(\mathbf{x})$, and debiased logits $\mathbf{s}_{\text{deb}}(\mathbf{x})$ for each test sample

3: **Warm-up:** sample $\{\mathbf{x}_i\}_{i=1}^{N_S}$; compute finite-difference responses by Eq. (3); form covariance by Eq. (4); extract $\widehat{\mathbf{V}} \in \mathbb{R}^{d\times r}$ and $\widehat{\mathbf{P}} = \widehat{\mathbf{V}}\widehat{\mathbf{V}}^\top$

4: **for** each test sample $\mathbf{x}$ **do**

5: $\quad \mathbf{f}(\mathbf{x}) \leftarrow \mathcal{N}(E_{\text{img}}(\mathbf{x}))$

6: $\quad \mathbf{z}_{\text{sty}}(\mathbf{x}) \leftarrow \widehat{\mathbf{V}}^\top \mathbf{f}(\mathbf{x})$

7: $\quad \mathbf{f}_{\text{cnt}}(\mathbf{x}) \leftarrow \mathcal{N}(\mathbf{f}(\mathbf{x}) - \lambda_{\text{proj}}\widehat{\mathbf{P}}\,\mathbf{f}(\mathbf{x}))$

8: $\quad$ route $k^*(\mathbf{x}) \leftarrow \arg\min_k \|\mathbf{z}_{\text{sty}}(\mathbf{x}) - \mathbf{c}_k\|_2^2$

9: $\quad$ update centroid $\mathbf{c}_{k^*(\mathbf{x})}$ by Eq. (9)

10: $\quad$ update bias template $\mathbf{b}^{(k^*(\mathbf{x}))}$ by Eq. (11)

11: $\quad \mathbf{s}_{\text{deb}}(\mathbf{x}) \leftarrow \mathbf{s}_{\text{clip}}(\mathbf{x}) - \gamma_{\text{env}}\mathbf{b}^{(k^*(\mathbf{x}))}$

12: $\quad$ output $\big(\mathbf{f}_{\text{cnt}}(\mathbf{x}), \mathbf{z}_{\text{sty}}(\mathbf{x}), \mathbf{s}_{\text{deb}}(\mathbf{x})\big)$

13: **end for**

---

## 3. Plug-and-Play Instantiations

### 3.1. Instantiation A: D²O+TDA (cache-based retrieval host)

A cache-based adapter typically augments the CLIP zero-shot logits $\mathbf{s}_{\text{clip}}(\mathbf{x})$ (Eq. (2)) with non-parametric evidence retrieved from a knowledge bank:

$$\mathbf{s}_{\text{cache}}(\mathbf{x}) = \mathbf{s}_{\text{clip}}(\mathbf{x}) + \Delta\mathbf{s}_{\text{pos}}(\mathbf{x}) - \Delta\mathbf{s}_{\text{neg}}(\mathbf{x}),$$

where $\Delta\mathbf{s}_{\text{pos}}$ aggregates support from retrieved *positive* neighbors, while $\Delta\mathbf{s}_{\text{neg}}$ provides a complementary *negative* correction term for suppressing spurious matches.

Building on this template, D²O+TDA plugs D²O into TDA in two places: (i) retrieval uses the retrieval-oriented content feature $\mathbf{f}_{\text{cnt}}(\mathbf{x})$ instead of the raw feature, yielding $\Delta\mathbf{s}_{\text{pos,cnt}}(\mathbf{x})$; and (ii) the CLIP prior is replaced by the debiased prior $\mathbf{s}_{\text{deb}}(\mathbf{x})$ from Eq. (12). We provide a schematic of this architecture in Appendix A (Figure 4), leading to the following formulation:

$$\mathbf{s}_{\text{D}^2\text{O+TDA}}(\mathbf{x}) = \mathbf{s}_{\text{deb}}(\mathbf{x}) + \Delta\mathbf{s}_{\text{pos,cnt}}(\mathbf{x}) - \Delta\mathbf{s}_{\text{neg}}(\mathbf{x}). \quad (13)$$

### 3.2. Instantiation B: D²O+ADAPT (Gaussian training-free posterior host)

We maintain a knowledge bank

$$\mathcal{B} = \{(\mathbf{f}_j, \mathbf{f}_{\text{cnt},j}, \hat{\mathbf{p}}_j)\}_{j=1}^{|\mathcal{B}|},$$

where $\mathbf{f}_j = \mathbf{f}(\mathbf{x}_j) \in \mathbb{R}^d$ denotes the raw feature used for Gaussian discriminant analysis (GDA) statistics, $\mathbf{f}_{\text{cnt},j} = \mathbf{f}_{\text{cnt}}(\mathbf{x}_j) \in \mathbb{R}^d$ is the retrieval-oriented content feature, and $\hat{\mathbf{p}}_j \in \Delta^{C-1}$ is the soft-label vector. The inference pipeline, highlighting this hybrid feature usage and branch-wise fusion, is visualized in Appendix A (Figure 5).

Building on the debiased logits from D²O, D²O+ADAPT defines an unnormalized class posterior score as

$$p(y{=}k \mid \mathbf{x}) \propto \underbrace{\mathcal{S}(\mathbf{s}_{\text{deb}}(\mathbf{x}))_k}_{\text{debiased prior}} \cdot \underbrace{\exp(\ell_k(\mathbf{x}))}_{\text{Gaussian likelihood}} \cdot \underbrace{\exp(g_k(\mathbf{x}; \mathcal{B}))}_{\text{bank evidence}}. \quad (14)$$

**Gaussian likelihood (Distribution Alignment).** For the query sample $\mathbf{x}$, let $\mathbf{f}(\mathbf{x}) \in \mathbb{R}^d$ denote its raw feature, whereas $\mathbf{f}_j$ denotes the raw feature stored in the bank. To preserve full distributional statistics for alignment (e.g., style-induced variance), we estimate class-conditional Gaussian parameters using raw features. Specifically, the class means $\boldsymbol{\mu}_k$ and shared covariance $\boldsymbol{\Sigma}$ are updated from $\mathcal{B}$ as

$$\boldsymbol{\mu}_k = \frac{\sum_{j\in\mathcal{B}} \hat{p}_{j,k}\, \mathbf{f}_j}{\sum_{j\in\mathcal{B}} \hat{p}_{j,k}}, \quad (15)$$

$$\boldsymbol{\Sigma} = \frac{\sum_{j\in\mathcal{B}} \sum_{k=1}^{C} \hat{p}_{j,k}\big(\mathbf{f}_j - \boldsymbol{\mu}_k\big)\big(\mathbf{f}_j - \boldsymbol{\mu}_k\big)^\top}{\sum_{j\in\mathcal{B}} \sum_{k=1}^{C} \hat{p}_{j,k}} + \eta\mathbf{I}, \quad (16)$$

where $\hat{p}_{j,k}$ denotes the class-$k$ component of the soft-label vector $\hat{\mathbf{p}}_j$, and $\eta > 0$ is a regularization term. The resulting likelihood is

$$\ell_k(\mathbf{x}) = -\tfrac{1}{2}\big(\mathbf{f}(\mathbf{x}) - \boldsymbol{\mu}_k\big)^\top \boldsymbol{\Sigma}^{-1}\big(\mathbf{f}(\mathbf{x}) - \boldsymbol{\mu}_k\big). \quad (17)$$

**Bank evidence (Semantic Retrieval).** To reduce retrieval confounding and emphasize semantic similarity, we compute similarity weights using the retrieval-oriented content features stored in the bank:

$$w_j(\mathbf{x}) = \frac{\exp\big(\mathbf{f}_{\text{cnt}}(\mathbf{x})^\top \mathbf{f}_{\text{cnt},j}/\tau_w\big)}{\sum_{m\in\mathcal{B}} \exp\big(\mathbf{f}_{\text{cnt}}(\mathbf{x})^\top \mathbf{f}_{\text{cnt},m}/\tau_w\big)}, \quad (18)$$

where $\tau_w > 0$ is a temperature parameter. We then aggregate the similarity-weighted votes as

$$g_k(\mathbf{x}; \mathcal{B}) = \sum_{j\in\mathcal{B}} w_j(\mathbf{x})\, \hat{p}_{j,k}. \quad (19)$$

**Extension to Transductive Protocol.** When the entire unlabeled test set $\mathcal{D} = \{\mathbf{x}_i\}_{i=1}^{N}$ is available in the transductive setting, we first apply D²O to all instances in a single pass, thereby obtaining debiased priors and retrieval-oriented content features for each test sample. Next, based on these debiased predictions, we construct a global knowledge bank $\mathcal{B}$ by selecting, for each class, the top-$L$ most confident samples from $\mathcal{D}$ as reliable support. Finally, leveraging both the soft statistics aggregated over $\mathcal{D}$ and the high-confidence evidence retained in $\mathcal{B}$, we estimate the class means via a one-pass closed-form solution:

$$\mu_k^{\text{trans}} = \frac{\sum_{i=1}^{N} \hat{p}_{i,k}\mathbf{f}_i + \sum_{j \in \mathcal{B}_k} \hat{p}_{j,k}\mathbf{f}_j}{\sum_{i=1}^{N} \hat{p}_{i,k} + \sum_{j \in \mathcal{B}_k} \hat{p}_{j,k}}, \qquad (20)$$

where $\hat{p}_{i,k}$ is the debiased zero-shot probability. For robustness, the shared covariance $\Sigma$ is estimated solely using the reliable samples in $\mathcal{B}$, and final predictions follow Eq. (14).

## 4. Experiments

### 4.1. Setup

**Datasets.** We evaluate D²O on two representative training-free host families, namely D²O+ADAPT and D²O+TDA, across three tasks: natural distribution shift, corruption robustness, and fine-grained categorization. **(i) Natural distribution shift.** We evaluate on the challenging Photorealistic Unreal Graphics (PUG) dataset (Bordes et al., 2023), which exhibits substantial variations in texture, background, scale, and orientation, making it a suitable benchmark for style/environment-dominant shifts. We additionally consider four out-of-distribution (OOD) variants of ImageNet (Deng et al., 2009): ImageNet-V2 (Recht et al., 2019), ImageNet-Sketch (Wang et al., 2019), ImageNet-A (Hendrycks et al., 2021b), and ImageNet-R (Hendrycks et al., 2021a). **(ii) Corruption robustness.** We conduct experiments on ImageNet-C, which contains 15 corruption types (e.g., Gaussian noise, motion blur, snow, and pixelation) grouped into four categories: noise, blur, weather, and digital. **(iii) Fine-grained categorization.** Following TPT, we evaluate on 10 image-classification datasets spanning diverse domains and difficulty levels, including Caltech101 (Fei-Fei et al., 2004), OxfordPets (Parkhi et al., 2012), StanfordCars (Krause et al., 2013), Flowers102 (Nilsback & Zisserman, 2008), Food101 (Bossard et al., 2014), FGVC-Aircraft (Maji et al., 2013), SUN397 (Xiao et al., 2010), DTD (Cimpoi et al., 2014), EuroSAT (Helber et al., 2019), and UCF101 (Soomro et al., 2012).

**Implementation Details.** We evaluate D²O under two inference-time protocols: *Online* and *Transductive*. Unless otherwise specified, we use CLIP (Radford et al., 2021) with ViT-B/16 as the visual backbone. For the D²O core parameters, we estimate the nuisance-sensitive basis $\hat{\mathbf{V}}$

using a warm-up buffer of $N_S = 128$ samples with subspace dimension $r = 8$, and set the projection strength to $\lambda_{\text{proj}} = 0.4$. For *environment debiasing*, we use $K_{\text{env}} = 16$ routing clusters, EMA momentum $\rho = 0.05$, and debiasing strength $\gamma_{\text{env}} = 0.8$. For *host-specific inherited parameters*, D²O+ADAPT uses a Gaussian mean-update momentum $\alpha = 0.9$. All experiments are conducted on a single NVIDIA RTX 5880 Ada Generation.

**(i) Online Protocol.** Our default online setting uses a short initialization/calibration buffer drawn from the same evaluation stream. After this initialization stage, test instances are processed sequentially (batch size = 1) without revisiting past samples. Unless otherwise specified, we use $L = 16$ samples per class in the knowledge bank.

**(ii) Transductive Protocol.** In the transductive setting, adaptation is performed with access to the full unlabeled test set. Unless otherwise specified, we set the knowledge bank size to $L = 6$ for stable global selection.

### 4.2. Main Results

**Task 1: Natural Distribution Shift. (i) PUG.** As reported in Table 1, D²O+ADAPT achieves 58.28% in the **Online** setting, outperforming ADAPT (51.87%) by 6.41%. The PUG setting is where D²O provides the clearest benefit. The gains are especially pronounced on appearance-dominated factors such as Lighting (43.88% vs. 27.47%) and Texture (56.78% vs. 51.06%), while also remaining strong on geometric factors such as Camera Yaw (71.52%). In the **Transductive** setting, D²O+ADAPT reaches 56.85%, again improving over ADAPT (52.85%). The slightly smaller gain in the transductive regime is reasonable, since PUG contains highly heterogeneous, instance-specific shifts that can benefit from sample-level correction. As a cross-host instantiation, D²O+TDA also improves over TDA in the online setting (52.22% vs. 51.50%), supporting the plug-and-play applicability of D²O beyond a single host family.

**(ii) ImageNet Variants.** Table 2 shows that D²O+ADAPT achieves the strongest average among the compared methods in our evaluation, reaching 65.92% **Online** and 66.53% **Transductive**. Compared with PUG, the gains on standard OOD benchmarks are smaller but still consistent. The most visible improvements appear on style-heavy variants such as ImageNet-Sketch (53.69%) and ImageNet-R (81.29%), which is consistent with the intended role of D²O under style/environment-induced confounding.

**Task 2: Corruption Robustness.** On ImageNet-C (Table 3), D²O+ADAPT attains 29.15% in the **Online** setting, exceeding ADAPT (28.56%) and TDA (27.84%). Here the gains are incremental rather than dramatic compared with PUG, but remain broadly consistent across corruption families. Such consistency is desirable for training-free TTA,

*Table 1.* Top-1 accuracy (%) comparison under geometric and photometric shifts (PUG). The best results in each setting (Online and Transductive) are marked in **bold**, and the second-best results are underlined. "BP-free" indicates whether the method avoids backpropagation at test time.

| | Method | BP-free | Camera | | | Pose | | | Scale | Texture | Lighting | Worlds | Avg. |
|---|---|---|---|---|---|---|---|---|---|---|---|---|---|
| | | | Yaw | Pitch | Roll | Yaw | Pitch | Roll | | | | | |
| **Online** | MaPLe+TPT (Khattak et al., 2023) | ✗ | 56.12 | 45.51 | 38.49 | 56.19 | 36.33 | 32.67 | 53.91 | 42.96 | 22.20 | 40.99 | 42.54 |
| | PromptAlign (Abdul Samadh et al., 2023) | ✗ | 57.83 | 46.75 | 39.78 | 57.50 | 37.13 | 34.28 | 55.87 | 44.72 | 22.87 | 42.34 | 43.91 |
| | DMN (Zhang et al., 2024d) | ✗ | 60.73 | 43.28 | 46.41 | 59.97 | 37.89 | 34.94 | 56.19 | 46.84 | 24.03 | 34.18 | 44.45 |
| | DPE (Zhang et al., 2024a) | ✗ | 41.50 | 31.08 | 27.73 | 39.59 | 24.44 | 23.43 | 39.08 | 27.44 | 4.51 | 22.63 | 28.14 |
| | MTA (Zanella & Ben Ayed, 2024) | ✓ | 41.90 | 33.33 | 29.74 | 39.76 | 24.56 | 23.41 | 40.52 | 28.41 | 6.59 | 32.22 | 30.04 |
| | ZERO (Farina et al., 2024) | ✓ | 45.44 | 36.05 | 33.63 | 43.72 | 27.60 | 26.61 | 44.50 | 31.69 | 15.17 | 39.49 | 34.39 |
| | AWT (Zhu et al., 2024b) | ✓ | 42.91 | 34.78 | 32.51 | 41.32 | 26.38 | 23.94 | 41.94 | 28.53 | 5.86 | 30.67 | 30.88 |
| | TDA (Karmanov et al., 2024) | ✓ | 63.85 | 51.00 | 50.81 | 63.41 | 42.46 | 39.92 | 62.48 | 51.44 | 35.87 | 53.75 | 51.50 |
| | ADAPT (Zhang et al., 2025) | ✓ | 67.13 | 52.69 | 52.23 | 66.23 | 41.78 | 40.35 | 65.12 | 51.06 | 27.47 | 54.60 | 51.87 |
| | D²O+TDA(ours) | ✓ | 65.42 | 51.93 | 50.81 | 64.02 | 43.00 | 40.94 | 63.07 | 52.33 | 36.35 | 54.35 | 52.22 (+0.72) |
| | D²O+ADAPT(ours) | ✓ | **71.52** | **56.84** | **58.29** | **70.99** | **48.25** | **47.03** | **69.71** | **56.78** | **43.88** | **59.55** | **58.28** (+6.41) |
| **Trans.** | TIMO (Li et al., 2025) | ✓ | 64.43 | 53.45 | 53.89 | 65.65 | 43.62 | **46.43** | 63.97 | 53.25 | 38.87 | 42.69 | 52.63 |
| | StatA (Zanella et al., 2025) | ✓ | 55.00 | 43.74 | 36.21 | 53.98 | 29.94 | 29.44 | 49.85 | 43.90 | 15.91 | 35.81 | 39.38 |
| | ADAPT (Zhang et al., 2025) | ✓ | 68.28 | 53.84 | 53.89 | 67.43 | 43.49 | 41.92 | 64.32 | 51.81 | 29.14 | 54.34 | 52.85 |
| | D²O+ADAPT(ours) | ✓ | **70.65** | **55.87** | **57.82** | **69.89** | **48.57** | 45.99 | **68.64** | **54.84** | **39.90** | **56.33** | **56.85** (+4.00) |

where noisy retrieval and biased priors can be amplified under severe low-level degradation. The advantage also persists in the **Transductive** setting (31.27% vs. 30.34% for ADAPT), suggesting that global statistical estimation can further stabilize prediction under corruption.

**Task 3: Fine-Grained Categorization.** Across 10 fine-grained datasets (Table 4), D²O+ADAPT achieves the best online average accuracy of 71.12% (vs. 70.76% for ADAPT and 70.30% for DMN), and further improves to 72.62% in the **Transductive** setting. Again, the gains are more modest than on PUG, but remain consistent. The improvement is particularly visible on datasets with substantial intra-class variation, such as OxfordPets (92.48%) and Flowers102 (76.74%), where background and appearance cues can otherwise distract recognition. Overall, these results support the view that D²O is most beneficial under style/environment-dominant shifts, while still providing smaller but consistent gains on more standard OOD and fine-grained settings.

### 4.3. Ablation Studies and Further Analysis

**Component Ablation.** Table 5 isolates two internal components of D²O+ADAPT: (i) prior debiasing and (ii) retrieval-oriented content evidence. The environment-debiased prior contributes the larger share of the improvement, especially on PUG-like shifts, indicating that environment-dependent logit distortion is a primary bottleneck under severe style/environment variation. By contrast, the direct accuracy gain from retrieval-oriented content evidence is smaller, but it acts as an important refinement by making retrieval more semantic and less dominated by style cues. When both components are enabled, the method achieves the strongest overall performance, indicating that the two effects are complementary rather than redundant.

**Qualitative Analysis of Retrieval Mechanism.** Although

the accuracy gain from retrieval-oriented content evidence alone is modest in Table 5, its contribution becomes clearer when examining retrieval purity. As shown in Figure 2, we measure *Retrieval Purity*, i.e., the proportion of retrieved support samples sharing the same class label as the query. Raw feature retrieval often suffers from "background matching" under severe geometric or environmental shifts, retrieving samples with similar pose or appearance but incorrect class identity. In contrast, our retrieval-oriented content feature improves purity consistently across all shift factors, with gains of **+2.5%** on Camera and **+2.2%** on Scale. This supports the claim that D²O mitigates retrieval confounding by preserving a more semantically coherent knowledge bank. Additional retrieval purity results on other benchmarks are reported in Appendix F.4.

**Hybrid Feature Strategy (Raw vs. Retrieval-oriented Content).** D²O+ADAPT adopts a hybrid design: the retrieval branch uses the retrieval-oriented content feature $\mathbf{f}_{\mathrm{cnt}}$, while the Gaussian posterior branch retains the raw feature $\mathbf{f}$. This choice reflects the fact that style-related directions are not uniformly harmful: suppressing nuisance directions is useful for retrieval, whereas raw distributional statistics can still be informative for Gaussian alignment. In Appendix F.6, we provide a more detailed empirical comparison. Table 9 shows that while $\mathbf{f}_{\mathrm{cnt}}$ can offer small benefits on adversarial shifts (e.g., ImageNet-A), raw features remain a more robust default on natural distribution shifts and fine-grained recognition. Therefore, we retain raw features in the Gaussian branch to preserve discriminative statistics while using $\mathbf{f}_{\mathrm{cnt}}$ in the retrieval branch, where style suppression is most directly beneficial.

**Hyperparameter Analysis.** We analyze the sensitivity of key hyperparameters on PUG-ImageNet under both Online and Transductive settings (Figure 3). For environment debiasing, accuracy improves steadily with the strength $\gamma_{\mathrm{env}}$

*Table 2.* Top-1 accuracy (%) comparison on natural distribution shifts. The best results in each setting (Online and Transductive) are marked in **bold**, and the second-best results are underlined. "BP-free" indicates whether the method avoids backpropagation at test time.

| | Method | BP-free | ImageNet-V2 | ImageNet-Sketch | ImageNet-A | ImageNet-R | OOD Avg. |
|---|---|---|---|---|---|---|---|
| Online | CLIP (Radford et al., 2021) | - | 60.89 | 46.12 | 47.79 | 73.99 | 57.20 |
| | Tip-Adapter (Zhang et al., 2022) | ✗ | 63.41 | 48.88 | 51.04 | 77.76 | 60.27 |
| | TPT (Shu et al., 2022) | ✗ | 63.45 | 47.97 | 54.77 | 77.06 | 60.81 |
| | MaPLe+TPT (Khattak et al., 2023) | ✗ | 64.36 | 47.41 | 57.80 | 77.83 | 61.85 |
| | DiffTPT (Feng et al., 2023) | ✗ | 65.10 | 46.80 | 55.68 | 75.00 | 60.65 |
| | PromptAlign (Abdul Samadh et al., 2023) | ✗ | 64.70 | 49.71 | 59.16 | 79.03 | 63.15 |
| | C-TPT (Yoon et al., 2024) | ✗ | 62.70 | 47.90 | 51.60 | 76.00 | 59.55 |
| | DMN (Zhang et al., 2024d) | ✗ | 65.17 | 53.20 | 58.28 | 78.55 | 63.80 |
| | DPE (Zhang et al., 2024a) | ✗ | **65.44** | 52.26 | 59.63 | 80.40 | 64.43 |
| | TPS (Sui et al., 2025) | ✗ | 63.80 | 49.57 | 59.21 | 77.49 | 62.52 |
| | DynaPrompt (Xiao et al., 2025) | ✗ | 64.67 | 48.22 | 56.17 | 78.17 | 61.81 |
| | B²TPT (Meng et al., 2025) | ✗ | 65.40 | 49.53 | 55.26 | 78.64 | 62.21 |
| | MTA (Zanella & Ben Ayed, 2024) | ✓ | 63.67 | 48.51 | 57.23 | 76.98 | 61.60 |
| | ZERO (Farina et al., 2024) | ✓ | 65.41 | 50.55 | 62.83 | 80.63 | 64.86 |
| | AWT (Zhu et al., 2024b) | ✓ | 65.15 | 51.60 | 60.33 | 80.64 | 64.43 |
| | RA-TTA (Lee et al., 2025) | ✓ | 64.16 | 50.83 | 59.21 | 79.68 | 63.47 |
| | BCA (Zhou et al., 2025) | ✓ | 64.90 | 50.87 | 61.14 | 80.72 | 64.41 |
| | TCA (Wang et al., 2025) | ✓ | 62.10 | 48.95 | 50.13 | 77.11 | 59.57 |
| | Dota (Han et al., 2025) | ✓ | 64.41 | 51.33 | 61.19 | 81.17 | 64.53 |
| | TDA (Karmanov et al., 2024) | ✓ | 64.62 | 50.76 | 60.06 | 80.50 | 63.99 |
| | ADAPT (Zhang et al., 2025) | ✓ | 64.64 | 53.13 | 63.32 | 80.66 | 65.44 |
| | D²O+TDA (ours) | ✓ | 64.72 | 51.21 | 60.35 | 80.71 | 64.25 (+0.26) |
| | D²O+ADAPT (ours) | ✓ | 64.86 | **53.69** | **63.83** | **81.29** | **65.92** (+0.48) |
| Trans. | GDA-CLIP (Wang et al., 2024) | ✓ | 55.67 | 34.32 | 19.72 | 55.30 | 41.25 |
| | TransCLIP (Zanella et al., 2024) | ✓ | 62.30 | 49.70 | 49.50 | 75.00 | 59.13 |
| | Frolic (Zhu et al., 2024a) | ✓ | 64.70 | 53.30 | 60.40 | 80.70 | 64.78 |
| | TIMO (Li et al., 2025) | ✓ | 56.40 | 35.96 | 22.06 | 58.47 | 43.22 |
| | ADAPT (Zhang et al., 2025) | ✓ | 65.59 | 53.87 | 63.77 | 80.64 | 65.97 |
| | D²O+ADAPT (ours) | ✓ | **66.18** | **54.24** | **64.67** | **81.01** | **66.53** (+0.56) |

*Table 3.* Top-1 accuracy (%) comparison on corruption robustness. The best results in each setting (Online and Transductive) are marked in **bold**, and the second-best results are underlined. "BP-free" indicates whether the method avoids backpropagation at test time.

| | Method | BP-free | Defo. | Glas. | Moti. | Zoom | Snow | Fros. | Fog | Brig. | Cont. | Elas. | Pix. | JPEG | Gauss. | Shot | Impu. | Avg. |
|---|---|---|---|---|---|---|---|---|---|---|---|---|---|---|---|---|---|---|
| | | | Blur | | | | Weather | | | | Digital | | | | Noise | | | |
| Online | CLIP (Radford et al., 2021) | - | 24.25 | 15.71 | 24.46 | 22.60 | 33.08 | 31.06 | 37.61 | 55.62 | 17.11 | 13.43 | 33.04 | 33.70 | 13.25 | 14.16 | 13.48 | 25.50 |
| | TPT (Shu et al., 2022) | ✗ | **27.56** | 15.48 | 26.16 | **26.94** | 36.74 | 34.28 | 39.38 | 60.22 | 16.96 | 15.64 | **40.74** | **37.90** | 10.64 | 11.94 | 10.92 | 27.43 |
| | DiffTPT (Feng et al., 2023) | ✗ | 25.63 | 16.96 | 26.74 | 25.40 | 35.99 | 34.57 | 39.83 | 59.01 | 17.32 | 17.16 | 38.43 | 35.47 | 12.97 | 13.60 | 13.21 | 27.49 |
| | DMN (Zhang et al., 2024d) | ✗ | 26.06 | 17.19 | 26.61 | 25.23 | 34.81 | 33.48 | 38.93 | 58.70 | 19.38 | 15.40 | 35.32 | 36.49 | 14.33 | 15.33 | 14.69 | 27.46 |
| | TDA (Karmanov et al., 2024) | ✓ | 24.85 | 16.27 | 24.41 | 25.94 | 36.28 | 35.11 | 40.82 | 56.45 | 31.71 | 19.19 | 40.17 | 35.80 | 9.48 | 11.18 | 9.95 | 27.84 |
| | ADAPT (Zhang et al., 2025) | ✓ | 26.30 | 18.01 | 27.31 | 25.54 | 36.19 | 34.67 | 40.96 | 60.29 | 19.95 | 16.09 | 37.44 | 37.22 | 15.76 | 16.84 | 15.90 | 28.56 |
| | D²O+TDA (ours) | ✓ | 25.12 | 16.47 | 24.78 | 26.16 | 36.65 | **35.32** | 41.12 | 56.69 | **31.81** | **19.64** | 40.30 | 36.20 | 9.65 | 11.40 | 10.11 | 28.09 (+0.25) |
| | D²O+ADAPT (ours) | ✓ | 26.94 | **18.35** | **27.84** | 26.01 | **37.00** | 35.23 | **41.63** | **60.56** | 20.38 | 16.92 | 38.04 | 37.60 | **16.51** | **17.45** | **16.79** | **29.15** (+0.59) |
| Trans. | ZLaP (Stojnić et al., 2024) | ✓ | 24.88 | 16.13 | 25.77 | 24.36 | 34.43 | 32.63 | 38.56 | 58.42 | 17.53 | 14.21 | 33.72 | 35.52 | 12.83 | 14.03 | 13.27 | 26.42 |
| | TransCLIP (Zanella et al., 2024) | ✓ | 25.35 | 16.40 | 25.53 | 23.22 | 34.58 | 32.47 | 39.65 | 59.04 | 17.72 | 14.76 | 35.22 | 35.53 | 14.82 | 16.11 | 15.60 | 27.07 |
| | StatA (Zanella et al., 2025) | ✓ | 20.23 | 13.29 | 20.38 | 18.84 | 31.30 | 29.80 | 34.58 | 54.79 | 11.24 | 11.80 | 26.31 | 33.20 | 9.58 | 10.52 | 10.12 | 22.40 |
| | ADAPT (Zhang et al., 2025) | ✓ | 28.01 | 19.77 | 29.00 | 27.37 | 38.06 | 36.43 | 42.42 | 61.24 | 21.96 | 18.41 | 38.91 | 38.22 | 17.71 | 18.80 | 18.82 | 30.34 |
| | D²O+ADAPT (ours) | ✓ | **28.85** | **20.61** | **29.87** | **28.32** | **39.13** | **37.50** | **43.73** | **61.71** | **22.43** | **20.05** | **39.89** | **38.69** | **18.75** | **19.81** | **19.68** | **31.27** (+0.93) |

in both settings and peaks at $\gamma_{\text{env}} = 0.8$ (Online: 58.28%), indicating that sufficiently strong prior correction is important under severe style/environment shift. Regarding the number of routing clusters, the Transductive setting is relatively stable, whereas the Online setting degrades when $K_{\text{env}} \geq 32$, likely because overly fine partitioning leads to sparse cluster statistics. We therefore adopt $K_{\text{env}} = 16$ as a robust default. $D^2O$ also shows relatively low sensitivity to the projection strength $\lambda_{\text{proj}}$ and the warm-up size $N_S$, and we set $N_S = 128$ for efficiency. Finally, the bank size $L$ in the Online setting shows a clear inverted-U pattern, peaking at $L = 16$. In the Transductive setting, performance is less sensitive to $L$; we adopt $L = 6$ in the main results to remain aligned with ADAPT for a fair comparison, even though slightly larger values can yield marginal additional gains.

## 4.4. Boundary and Scope Evidence

We further add two forms of evidence to clarify the operating boundary and plug-and-play scope of $D^2O$. First, we construct mixed-style online streams by explicitly interleaving two different PUG subsets in a single stream. This setting stresses whether a single global correction is sufficient when multiple style regimes alternate over time. Second, we evaluate $D^2O$ on additional host families beyond the two main instantiations. These results are not intended as exhaustive coverage, but rather as supporting breadth evidence that $D^2O$ is an upstream debiasing layer rather than a host-specific modification across adapters.

**Mixed-style stream boundary check.** Table 6 reports the mixed-style boundary check. Compared with a single global EMA bias (**G-EMA**), the routed design (**Main**) is consis-

*Table 4.* Top-1 accuracy (%) comparison on fine-grained categorization. The best results in each setting (Online and Transductive) are marked in **bold**, and the second-best results are underlined. "BP-free" indicates whether the method avoids backpropagation at test time.

| | Method | BP-free | Aircraft | Caltech | Cars | DTD | EuroSAT | Flower | Food101 | Pets | Sun397 | UCF101 | Avg. |
|---|---|---|---|---|---|---|---|---|---|---|---|---|---|
| Online | CLIP (Radford et al., 2021) | - | 23.70 | 92.98 | 65.24 | 44.44 | 41.42 | 67.28 | 83.80 | 87.98 | 62.55 | 65.08 | 63.45 |
| | TPT (Shu et al., 2022) | ✗ | 24.78 | 94.16 | 66.87 | 47.75 | 42.44 | 68.98 | 84.67 | 87.79 | 65.50 | 68.04 | 65.10 |
| | MaPLe+TPT (Khattak et al., 2023) | ✗ | 24.54 | 94.00 | 66.83 | 46.49 | 48.06 | 72.23 | 86.94 | 90.65 | 67.65 | 69.44 | 66.68 |
| | DiffTPT (Feng et al., 2023) | ✗ | 25.60 | 92.49 | 67.01 | 47.00 | 43.13 | 70.10 | 87.23 | 88.22 | 65.74 | 68.22 | 65.47 |
| | PromptAlign (Abdul Samadh et al., 2023) | ✗ | 25.65 | 94.36 | 67.91 | 48.11 | 46.38 | 72.31 | 87.33 | 90.57 | 68.36 | 70.10 | 67.11 |
| | C-TPT (Yoon et al., 2024) | ✗ | 24.00 | 93.60 | 65.80 | 46.00 | 43.20 | 79.80 | 83.70 | 88.20 | 64.80 | 65.70 | 65.48 |
| | DMN (Zhang et al., 2024d) | ✗ | **30.03** | **95.38** | 67.96 | **55.85** | 59.43 | 74.49 | 85.08 | 92.04 | 70.18 | **72.51** | 70.30 |
| | TPS (Sui et al., 2025) | ✗ | 26.27 | 94.56 | 67.00 | 53.80 | 42.11 | 71.69 | 84.78 | 87.82 | 68.25 | 71.18 | 66.75 |
| | DPE (Zhang et al., 2024a) | ✗ | 28.95 | 94.81 | 67.31 | 54.20 | 55.79 | 75.07 | 86.17 | 91.14 | 70.07 | 70.44 | 69.40 |
| | HisTPT (Zhang et al., 2024c) | ✗ | 26.90 | 94.50 | 69.20 | 48.90 | 49.70 | 71.20 | 89.30 | 89.10 | 67.20 | 70.10 | 67.61 |
| | DynaPrompt (Xiao et al., 2025) | ✗ | 24.33 | 94.32 | 67.65 | 47.96 | 42.28 | 69.95 | 85.42 | 88.28 | 66.32 | 68.72 | 65.52 |
| | MTA (Zanella & Ben Ayed, 2024) | ✓ | 27.69 | 95.21 | 65.60 | 52.90 | 55.43 | 73.85 | 83.99 | 90.62 | 67.97 | 67.96 | 68.12 |
| | ZLaP (Stojnić et al., 2024) | ✓ | 25.40 | 93.10 | 65.60 | 48.60 | 55.60 | 73.50 | 86.90 | 87.10 | 67.40 | 71.50 | 67.47 |
| | ZERO (Farina et al., 2024) | ✓ | 27.66 | 95.21 | 65.63 | 52.90 | 55.51 | 73.85 | 84.00 | 90.57 | 67.95 | 67.96 | 68.12 |
| | BCA (Zhou et al., 2025) | ✓ | 28.59 | 94.69 | 66.86 | 53.49 | 56.63 | 73.12 | 85.97 | 90.43 | 68.41 | 67.59 | 68.58 |
| | OGA (Fuchs et al., 2025) | ✓ | 23.20 | 93.60 | 68.10 | 47.90 | 54.20 | 69.20 | 85.60 | 89.40 | 67.90 | 67.40 | 67.05 |
| | TCA (Wang et al., 2025) | ✓ | 24.87 | 93.63 | 65.33 | 46.16 | **70.43** | 73.33 | 85.31 | 89.53 | 65.92 | 72.38 | 68.69 |
| | Dota (Han et al., 2025) | ✓ | 25.59 | 94.32 | 69.48 | 47.87 | 57.65 | 74.67 | 87.02 | 91.69 | 69.70 | 72.06 | 69.01 |
| | TDA (Karmanov et al., 2024) | ✓ | 25.11 | 94.73 | 67.07 | 45.09 | 62.36 | 71.82 | 86.13 | 89.67 | 67.62 | 71.24 | 68.08 |
| | ADAPT (Zhang et al., 2025) | ✓ | 28.95 | 94.48 | 68.19 | 55.20 | 68.19 | 75.56 | 83.81 | 92.01 | 70.57 | 70.66 | 70.76 |
| | D²O+TDA (ours) | ✓ | 24.96 | 94.56 | 67.59 | 46.39 | 65.22 | 71.54 | 86.20 | 90.08 | 67.86 | 71.32 | 68.57 (+0.49) |
| | D²O+ADAPT (ours) | ✓ | 29.10 | 95.13 | 68.91 | 55.02 | 67.86 | 76.74 | 83.90 | **92.48** | 70.56 | 71.45 | **71.12** (+0.36) |
| Trans. | GDA-CLIP (Wang et al., 2024) | ✓ | 18.69 | 87.53 | 60.78 | 46.81 | 49.92 | 72.65 | 78.25 | 89.90 | 63.60 | 68.70 | 63.68 |
| | ZLaP (Stojnić et al., 2024) | ✓ | 26.30 | 91.80 | 66.80 | 46.00 | 57.70 | 67.90 | 87.20 | 87.90 | 67.80 | 73.80 | 67.32 |
| | TransCLIP (Zanella et al., 2024) | ✓ | 26.90 | 92.70 | 69.40 | 49.50 | 65.10 | 76.70 | 87.10 | 92.60 | 68.90 | 74.40 | 70.33 |
| | Frolic (Zhu et al., 2024a) | ✓ | **31.40** | 95.10 | 69.10 | 56.10 | 58.50 | 74.80 | 87.10 | **92.90** | 70.80 | **75.20** | 71.10 |
| | StatA (Zanella et al., 2025) | ✓ | 24.70 | 94.20 | 68.00 | 48.40 | **67.30** | 75.20 | 87.10 | 92.40 | 68.70 | 73.50 | 69.95 |
| | ADAPT (Zhang et al., 2025) | ✓ | 30.81 | 95.46 | 71.32 | 56.86 | 65.93 | 80.11 | 85.15 | 92.59 | 72.25 | 73.86 | 72.43 |
| | D²O+ADAPT (ours) | ✓ | 30.78 | **95.98** | 71.40 | **56.91** | 66.36 | **80.55** | 85.17 | 92.61 | 72.40 | 74.02 | **72.62** (+0.19) |

*Table 5.* **Component ablation of $D^2O$+ADAPT in the online setting.** We isolate (i) prior debiasing and (ii) retrieval-oriented content evidence. We additionally report Top-5 retrieval purity (Pur@5) to reflect the quality of semantic retrieval.

| Method | Ablations | | Performance (higher is better) | | | | | |
|---|---|---|---|---|---|---|---|---|
| | Prior debias | Content evid. | Task 1: PUG+IN | | Task 2: Corr. robust. | | Task 3: Fine-grained | |
| | | | Acc. | Pur@5 | Acc. | Pur@5 | Acc. | Pur@5 |
| Baseline (w/o both) | ✗ | ✗ | 58.66 | – | 28.56 | – | 70.76 | – |
| Ablation 1: + Content evid. | ✗ | ✓ | 58.70 | 53.22 | 28.52 | 22.31 | 70.83 | 64.21 |
| Ablation 2: + Prior debias | ✓ | ✗ | 62.01 | 51.69 | 29.15 | 22.31 | 71.10 | 62.46 |
| Full (D²O+ADAPT) | ✓ | ✓ | 62.10 | 53.22 | 29.15 | 22.31 | 71.12 | 64.21 |

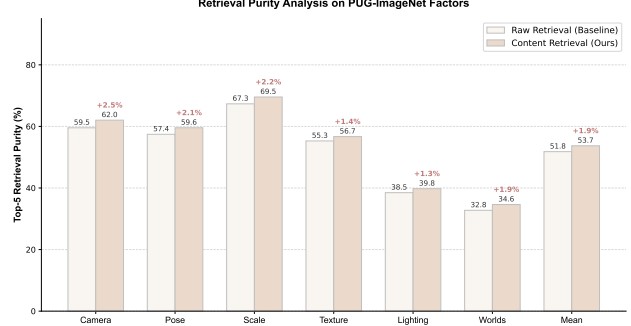

**Retrieval Purity Analysis on PUG-ImageNet Factors**

*Figure 2.* **Impact of retrieval-oriented content features on retrieval purity.** We analyze the purity of the retrieved Top-5 support samples, i.e., the proportion of retrieved samples matching the ground-truth class, across PUG-ImageNet factors. Although the direct accuracy gain from retrieval-oriented content evidence is modest in Table 5, the retrieval-oriented content feature consistently improves semantic consistency over raw retrieval, with notable gains on geometric factors such as **+2.5%** on Camera. This supports the view that the retrieval branch of $D^2O$ acts as a semantic safeguard against style-confounded retrieval.

tently stronger on all three interleaved streams, with an average advantage of +4.78. It also remains above raw feature clustering (**Raw**) on all three streams, although the margin is smaller (+1.19 on average). This supports the necessity of routing-based bias correction when heterogeneous style regimes coexist in the same online stream.

*Table 6.* **Mixed-style stream boundary check.** We explicitly interleave two different PUG subsets in one online stream and compare the routed design (**Main**) against a single global EMA bias (**G-EMA**) and raw feature clustering (**Raw**). Numbers in parentheses in the average row denote the gap to **Main**.

| Pair | CLIP | Main | G-EMA | Raw |
|---|---|---|---|---|
| cyaw + slight | 42.27 | **48.61** | 44.24 | 48.42 |
| croll + otexture | 47.91 | **52.56** | 48.35 | 49.59 |
| opitch + slight | 31.18 | **41.51** | 35.75 | 41.11 |
| Avg. | 40.45 | **47.56** | 42.78 (-4.78) | 46.37 (-1.19) |

**Scope extension to additional host families.** We also test whether $D^2O$ can be inserted into additional adaptation families. Table 7 summarizes breadth checks on CoOp, MTA, ZERO, and TPT. CoOp denotes the official ImageNet-trained evaluation-only checkpoint transferred to the target benchmarks. The strongest gains appear on PUG-like shifts for CoOp and TPT, while MTA and ZERO show milder but mostly positive improvements, especially on Fine-Grained-10. We therefore interpret these results as scope-support evidence rather than as the main empirical claim.

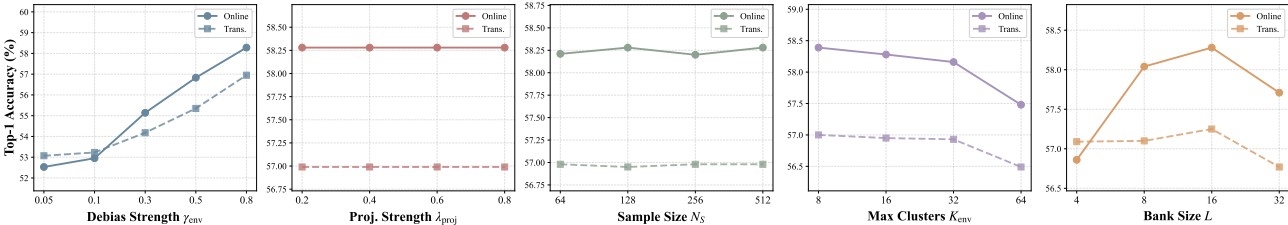

*Figure 3.* **Hyperparameter sensitivity analysis on PUG-ImageNet.** We report Top-1 Accuracy (%) while varying (a) Environment debias strength $\gamma_{\text{env}}$, (b) Projection strength $\lambda_{\text{proj}}$, (c) Sample size for style–nuisance estimation $N_S$, (d) Max environment clusters $K_{\text{env}}$, and (e) knowledge bank size $L$.

*Table 7.* **Scope extension to additional host families.** We report average accuracies before and after inserting D²O. CoOp denotes the official ImageNet-trained evaluation-only checkpoint.

| Host | PUG | | | FG-10 | | | IN-4 | | |
|------|------|-------|------|-------|-------|------|-------|-------|------|
| | Base | +D²O | Δ | Base | +D²O | Δ | Base | +D²O | Δ |
| CoOp | 31.08 | **36.88** | +5.80 | 61.77 | **64.71** | +2.94 | 58.50 | **60.07** | +1.57 |
| MTA | 30.04 | **30.38** | +0.34 | 68.12 | **70.81** | +2.69 | 61.60 | **62.61** | +1.01 |
| ZERO | 34.39 | **34.93** | +0.54 | 68.12 | **70.83** | +2.71 | 64.86 | **64.98** | +0.12 |
| TPT | 27.42 | **33.66** | +6.24 | 65.10 | **65.71** | +0.61 | 60.81 | **63.61** | +2.80 |

*Table 8.* **Efficiency comparison on ImageNet-Sketch.** We report Top-1 accuracy (%), gain relative to CLIP (%), total inference time, and peak GPU memory usage on the full ImageNet-Sketch validation set (50,889 images) using a single NVIDIA RTX 5880 Ada Generation. CLIP is included as a zero-shot reference; boldface marks the best adapted result within each protocol, with lower values preferred for time and memory.

| Setting | Method | BP-free | Acc. (%) ↑ | Gain (%) ↑ | Time ↓ | Mem. (GB) ↓ |
|---------|--------|---------|-----------|-----------|--------|-------------|
| Online | CLIP (Radford et al., 2021) | - | 45.95 | - | 2m 40s | 2.91 |
| | TPT (Shu et al., 2022) | × | 47.95 | +2.00 | 8h 54m | 18.21 |
| | TDA (Karmanov et al., 2024) | ✓ | 50.83 | +4.88 | 2h 22m | **0.69** |
| | ADAPT (Zhang et al., 2025) | ✓ | 53.13 | +7.18 | 3h 04m | 0.92 |
| | D²O+ADAPT (ours) | ✓ | **53.69** | **+7.74** | **2h 03m** | 1.04 |
| Trans. | ADAPT (Zhang et al., 2025) | ✓ | 53.87 | +7.92 | **57m 39s** | 28.46 |
| | D²O+ADAPT (ours) | ✓ | **54.24** | **+8.29** | 1h 07m | 28.46 |

Taken together, these results support two useful conclusions. First, the mixed-style experiment shows a boundary case where routing is more reliable than a single global bias. Second, the extra-family experiments suggest that D²O transfers beyond the two main-paper hosts, although improvements still vary across host families and benchmarks.

**Cost Comparison.** We evaluate computational cost on ImageNet-Sketch (Table 8). In the online setting, D²O+ADAPT substantially outperforms gradient-based TPT in efficiency, offering about $4\times$ faster inference and roughly $17\times$ lower memory usage. Compared with the training-free baseline ADAPT, it also achieves higher accuracy (53.69%) with an approximately 33% speedup. This efficiency comes from the lightweight closed-form projection and the stable inference objects produced by D²O, which promote faster bank convergence without requiring test-time backpropagation. With a memory footprint of 1.04 GB, the method remains practical for deployment in resource-constrained training-free settings.

In the transductive setting, D²O+ADAPT has the same memory footprint as ADAPT. This large memory usage is inherent to the transductive protocol itself, which requires storing features for the entire evaluation set, rather than being an additional overhead introduced by D²O.

*Remark* 4.1. Additional experimental results are reported in Appendix F, including experiments with different backbones, multiple random seeds, detailed boundary/scope evidence, and further mechanism analyses.

## 5. Limitations and Conclusion

**Limitations.** Our D²O-based approach has three limitations. **(i) Transductive memory overhead.** D²O+ADAPT requires storing full-test-set features for global statistics, which may be costly on memory-constrained devices. **(ii) Low-rank style–nuisance assumption.** D²O assumes dominant style/environment variation can be approximated by a low-rank finite-difference subspace; under high-rank, rapidly changing, or adversarial shifts, this estimate may be less reliable and may suppress useful semantics. **(iii) Fixed routing capacity.** Online routing uses a fixed number of clusters $K_{\text{env}}$, which may be insufficient for highly imbalanced or complex mixed-style streams. These limitations motivate adaptive routing, more flexible subspace estimation, and memory-efficient transductive variants.

**Conclusion.** We presented D²O, a training-free debiasing operator for test-time adaptation under style/environment shift. Rather than aiming at perfect style–content disentanglement, D²O constructs an inference-oriented decomposition into a style-aware routing coordinate, a retrieval-oriented content feature, and debiased logits. When instantiated in retrieval-based and Gaussian posterior hosts, D²O achieves its clearest improvements under style/environment-dominant shifts, with smaller but consistent gains on standard OOD, corruption, and fine-grained benchmarks. Our analysis connects operator-level estimation errors to decision-level robustness, supporting D²O as a practical inference-time debiasing layer.

## Acknowledgements

The work is supported by the National Natural Science Foundation of China (62125111, 62476268, 62206273), the Guangdong Provincial Key Laboratory of Multimodality Non-Invasive Brain-Computer Interface (Grant No.2024B1212010010), the Shenzhen Science and Technology Program (Grant No.JCYJ20240813155840052) and the Natural Science Foundation of Fujian Province of China (Grant No.2024J01158).

## Impact Statement

This paper presents work whose goal is to advance the field of machine learning. There are many potential societal consequences of our work, none of which we feel must be specifically highlighted here.

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

# Appendix

## A. Framework Overviews

In this section, we provide schematic illustrations of the two plug-and-play host instantiations: **D²O+TDA** (Figure 4) and **D²O+ADAPT** (Figure 5). These diagrams show the complete inference pipeline and how D²O maps each test image to three inference-time objects: a retrieval-oriented content feature $\mathbf{f}_{\mathrm{cnt}}(\mathbf{x})$, a style-aware routing coordinate $\mathbf{z}_{\mathrm{sty}}(\mathbf{x})$, and a debiased logit vector $\mathbf{s}_{\mathrm{deb}}(\mathbf{x})$. They also clarify their distinct roles: $\mathbf{z}_{\mathrm{sty}}(\mathbf{x})$ is used for style-aware environment routing and bias tracking, $\mathbf{f}_{\mathrm{cnt}}(\mathbf{x})$ drives semantic retrieval in the cache/knowledge bank, while the raw feature $\mathbf{f}(\mathbf{x})$ in D²O+ADAPT is retained for global statistical estimation.

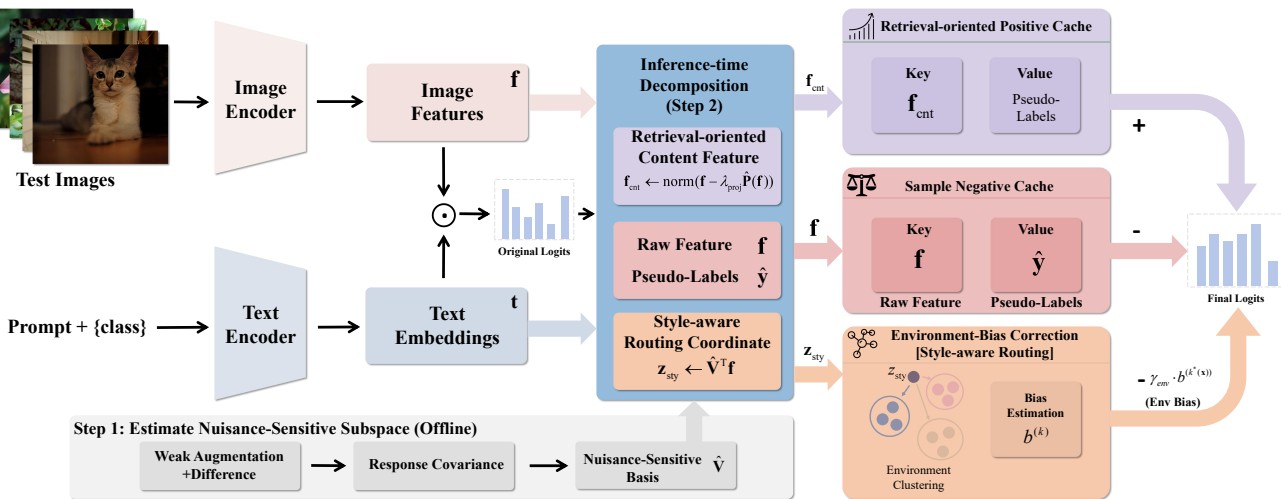

*Figure 4.* **Detailed framework of D²O+TDA (Instantiation A).** This architecture integrates the D²O operator into a retrieval-based adaptation pipeline. The process begins with an offline estimation of the nuisance-sensitive basis $\hat{\mathbf{V}}$. During inference, the decomposition module produces three inference-time objects: a retrieval-oriented content feature $\mathbf{f}_{\mathrm{cnt}}(\mathbf{x})$, a style-aware routing coordinate $\mathbf{z}_{\mathrm{sty}}(\mathbf{x})$, and the corresponding prediction signal used for downstream adaptation. The pipeline then combines three branches: (1) the **Retrieval-oriented Positive Cache**, which retrieves support samples using $\mathbf{f}_{\mathrm{cnt}}(\mathbf{x})$ to improve semantic consistency; (2) the **Sample Negative Cache**, which uses raw features $\mathbf{f}(\mathbf{x})$ to suppress dissimilar categories; and (3) the **environment-aware bias correction branch**, which uses the style-aware routing coordinate $\mathbf{z}_{\mathrm{sty}}(\mathbf{x})$ to identify the routed environment cluster and subtract the estimated centered-logit bias template $\mathbf{b}^{(k)}$ from the CLIP logits before producing the final prediction.

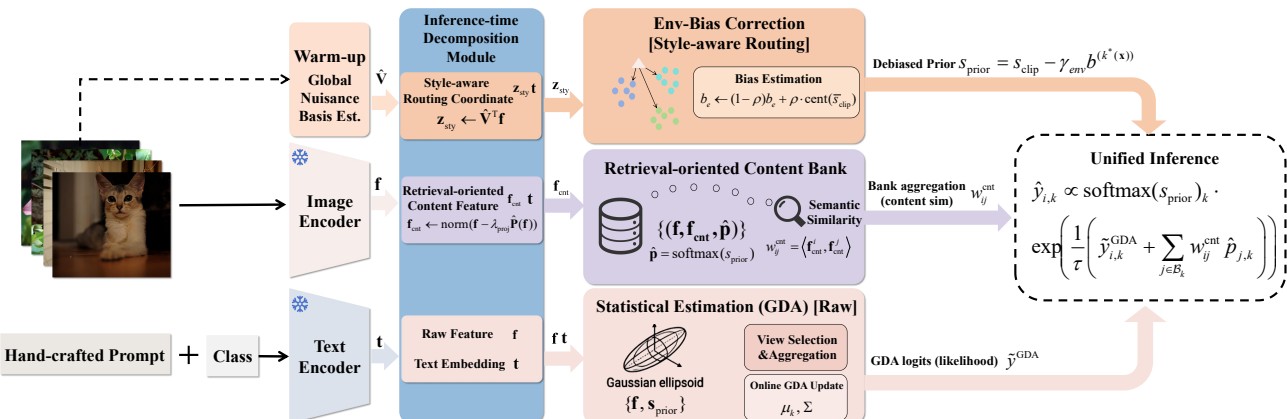

*Figure 5.* **Detailed framework of D²O+ADAPT (Instantiation B).** This framework combines the D²O operator with a probabilistic Gaussian posterior. The inference pipeline contains three parallel branches: (1) **environment-bias correction**, where the style-aware routing coordinate $\mathbf{z}_{\mathrm{sty}}(\mathbf{x})$ assigns the sample to an environment cluster and updates the centered-logit bias template $\mathbf{b}^{(k)}$, producing the debiased prior $S_{\mathrm{prior}}$; (2) **retrieval-oriented content bank**, where high-confidence samples are stored as tuples $(\mathbf{f}(\mathbf{x}), \mathbf{f}_{\mathrm{cnt}}(\mathbf{x}), \hat{\mathbf{p}})$ and retrieval weights $w_{ij}^{\mathrm{cnt}}$ are computed from semantic similarity in the retrieval-oriented content feature space; and (3) **statistical estimation (GDA)**, where the class means $\mu_k$ and covariance $\Sigma$ are estimated using raw features $\mathbf{f}(\mathbf{x})$ to preserve global distributional structure. The final prediction is obtained by fusing the debiased prior, the Gaussian likelihood, and the bank evidence.

# B. Related Work

**Online Test-Time Adaptation of VLMs.** In the online setting, test instances are processed sequentially, requiring the model to adapt continuously to the incoming stream (Wang et al., 2021). Early VLM-oriented online TTA methods mainly focus on gradient-based prompt tuning. Representative methods such as TPT (Shu et al., 2022), DiffTPT (Feng et al., 2023), and PromptAlign (Abdul Samadh et al., 2023), together with later variants such as DeYO (Lee et al., 2024) and DynaPrompt (Xiao et al., 2025), adapt the model by minimizing prediction entropy or related auxiliary objectives. While effective, these methods rely on iterative backpropagation at test time and can therefore incur substantial latency.

To improve efficiency, a line of training-free methods has emerged. Methods such as TDA (Karmanov et al., 2024), MTA (Zanella & Ben Ayed, 2024), and RA-TTA (Lee et al., 2025) refine predictions by retrieving high-confidence support samples from a cache or knowledge bank. More recently, ADAPT (Zhang et al., 2025) extends this direction by modeling class-conditional distributions with a closed-form Gaussian alignment module. However, these methods typically perform retrieval or distribution estimation directly in the original feature space, where semantic structure and nuisance/style cues are not explicitly separated. As a result, retrieved samples or estimated statistics can be biased by superficial appearance or environment cues rather than semantic identity.

Our work is complementary to these methods. Rather than proposing yet another host-specific online TTA family, we introduce D²O as an *operator-level debiasing layer* that can be inserted into existing training-free pipelines. Its role is to construct inference-oriented style–content signals for downstream adaptation: a style-aware routing coordinate, a retrieval-oriented content feature, and a debiased prior, so that semantic retrieval and bias correction become less sensitive to nuisance/style confounding.

**Transductive Test-Time Adaptation.** Unlike the online setting, the transductive setting permits access to the full unlabeled test set, making it possible to exploit global distribution structure. Some methods use graph-based propagation, such as ZLaP (Stojnić et al., 2024), to refine pseudo-labels. Others, including TransCLIP (Zanella et al., 2024), Frolic (Zhu et al., 2024a), TIMO (Li et al., 2025), StatA (Zanella et al., 2025), and the transductive version of ADAPT (Zhang et al., 2025), estimate global class-conditional structure to rectify predictions.

Although effective, many transductive methods rely on iterative optimization, graph construction, or other procedures that can become computationally expensive at scale. ADAPT provides a notably efficient closed-form alternative, but its estimation is still performed on features that may mix semantic and nuisance variation. In contrast, D²O serves as a lightweight inference-time debiasing operator before downstream transductive estimation: it uses retrieval-oriented content features when nuisance suppression benefits retrieval, uses style-aware routing coordinates for environment-aware bias tracking, and remains compatible with closed-form global alignment. This design allows us to improve robustness under style/environment shift while preserving the efficiency advantages of training-free transductive adaptation.

# C. Theoretical Analysis

**Purpose and roadmap.** This section formalizes why the *operator outputs* produced by D²O in Section 2—the retrieval-oriented content feature $\mathbf{f}_{\mathrm{cnt}}(\mathbf{x})$, the style-aware routing coordinate $\mathbf{z}_{\mathrm{sty}}(\mathbf{x})$, and the debiased logits $\mathbf{s}_{\mathrm{deb}}(\mathbf{x})$—are the *right inference objects* for robust training-free TTA. We prove guarantees at three levels: (i) *subspace identification* during warm-up, (ii) *online bias estimation* through routing and EMA, and (iii) *decision stability* for a downstream host adapter, namely D²O+ADAPT.

**Oracle objects.** Throughout, quantities marked with $\star$ denote oracle quantities computed using the true nuisance-sensitive subspace $\mathbf{V}_\star$ and the true centered environment bias $\mathbf{b}_e$. Our bounds quantify how estimation errors in $(\widehat{\mathbf{V}}, \widehat{\mathbf{b}}_e)$ propagate to bounded perturbations of decision statistics.

## C.1. Models

*Definition* C.1 (Confounded feature model). Let $y$ be the label and $e$ the environment. Assume the normalized feature satisfies

$$\mathbf{f} = \mathbf{u}_y + \mathbf{V}_\star \mathbf{z}_e + \varepsilon, \tag{21}$$

where $\mathbf{V}_\star \in \mathbb{R}^{d \times r}$ has orthonormal columns and represents the true nuisance-sensitive subspace, and $\mathbf{V}_\star^\top \mathbf{u}_y = \mathbf{0}$. (**We empirically verify this orthogonality condition in Appendix F.8, Figure 11(b).)**

*Definition* C.2 (Centered environment logit bias). Assume there exist centered bias vectors $\mathbf{b}_e \in \mathbb{R}^C$ ($\mathbf{1}^\top \mathbf{b}_e = 0$) such that

$$\mathcal{C}(\mathbf{s}_{\text{clip}}(\mathbf{x})) = \mathcal{C}(\mathbf{s}_\star(\mathbf{x})) + \mathbf{b}_{e(\mathbf{x})} + \boldsymbol{\xi}, \qquad \|\boldsymbol{\xi}\|_\infty \leq \sigma. \tag{22}$$

## C.2. Subspace recovery from finite differences

**Assumption (spiked response covariance).** Assume the finite-difference responses are sub-Gaussian with parameter $\kappa$ and satisfy $\mathbb{E}[\boldsymbol{\delta}\boldsymbol{\delta}^\top] = \mathbf{V}_\star \boldsymbol{\Lambda} \mathbf{V}_\star^\top + \sigma_\delta^2 \mathbf{I}$, where $\boldsymbol{\Lambda} = \text{diag}(\lambda_1, \ldots, \lambda_r)$ and $\lambda_r > 0$. **(See Appendix F.8, Figure 11(a) for empirical validation of this spectral structure.)**

*Theorem* C.3 (Nuisance-sensitive subspace recovery). *Let $\widehat{\mathbf{V}}$ be the top-$r$ eigenvectors of the covariance estimated from $N_\delta = N_S A$ finite-difference responses. Let $\Delta := \lambda_r$ be the eigengap. Then with probability at least $1 - \delta$,*

$$\|\sin\Theta(\widehat{\mathbf{V}}, \mathbf{V}_\star)\|_2 \leq \frac{c\,\kappa^2}{\Delta}\left(\sqrt{\frac{d + \log(1/\delta)}{N_\delta}} + \frac{d + \log(1/\delta)}{N_\delta}\right). \tag{23}$$

## C.3. Bias estimation by routing-based EMA

*Theorem* C.4 (EMA bias control under routing mistakes). *Let $\widehat{\mathbf{b}}_e$ denote the EMA template (step size $\rho$) for environment $e$ after $n$ updates. Assume centered-logit vectors $\|\mathcal{C}(\mathbf{s}_{\text{clip}}(\mathbf{x}))\|_\infty \leq B$ and routing error probability $p_{\text{rt}}$. With probability at least $1 - \delta$,*

$$\left\|\widehat{\mathbf{b}}_e - \mathbf{b}_e\right\|_\infty \leq (1-\rho)^n \left\|\widehat{\mathbf{b}}_e^{(0)} - \mathbf{b}_e\right\|_\infty + 2B\,p_{\text{rt}} + B\sqrt{\frac{2\log(2C/\delta)}{n_{\text{eff}}}}, \tag{24}$$

*where $n_{\text{eff}} \approx \frac{2-\rho}{\rho}$ is the effective sample size.*

## C.4. Decision-level robustness

**Posterior log-odds.** D²O+ADAPT uses the score factorization in Eq. (14). The pairwise log-odds $\Delta_{k\to y}(\mathbf{x}) = \log\frac{\tilde{p}_y(\mathbf{x})}{\tilde{p}_k(\mathbf{x})}$ decompose as

$$\Delta_{k\to y}(\mathbf{x}) = \underbrace{(\mathbf{s}_{\text{deb}}(\mathbf{x})_y - \mathbf{s}_{\text{deb}}(\mathbf{x})_k)}_{\text{Prior}} + \underbrace{\frac{1}{s}\left(\log P_{\text{gda}}(y \mid \mathbf{x}) - \log P_{\text{gda}}(k \mid \mathbf{x})\right)}_{\text{Gaussian}} + \underbrace{\beta\left(g_y(\mathbf{x};\mathcal{B}) - g_k(\mathbf{x};\mathcal{B})\right)}_{\text{Bank}}. \tag{25}$$

Here $\tilde{p}$ denotes the unnormalized posterior score, $s > 0$ is the Gaussian likelihood scale, and $\beta \geq 0$ is the bank evidence weight.

*Lemma* C.5 (Softmax stability). *For $\mathbf{w} = \mathcal{S}(\mathbf{a}/\tau)$ and $\mathbf{w}' = \mathcal{S}(\mathbf{a}'/\tau)$, it holds that $\|\mathbf{w} - \mathbf{w}'\|_1 \leq \frac{2}{\tau}\|\mathbf{a} - \mathbf{a}'\|_\infty$.*

*Lemma* C.6 (Quadratic discriminant perturbation). *Let $q(\mathbf{x};\boldsymbol{\mu},\boldsymbol{\Sigma}) = (\mathbf{x} - \boldsymbol{\mu})^\top \boldsymbol{\Sigma}^{-1}(\mathbf{x} - \boldsymbol{\mu})$. Under bounded inputs $R$ and condition numbers $M_\Sigma$,*

$$\left|q(\mathbf{x};\boldsymbol{\mu},\boldsymbol{\Sigma}) - q(\mathbf{x}';\boldsymbol{\mu}',\boldsymbol{\Sigma}')\right| \leq 4M_\Sigma R\left(\|\mathbf{x} - \mathbf{x}'\|_2 + \|\boldsymbol{\mu} - \boldsymbol{\mu}'\|_2\right) + 4M_\Sigma^2 R^2 \|\boldsymbol{\Sigma} - \boldsymbol{\Sigma}'\|_{\text{op}}. \tag{26}$$

*Lemma* C.7 (Inner-product perturbation). *Let $\mathbf{a}, \mathbf{a}', \mathbf{b}, \mathbf{b}'$ be vectors with unit $\|\cdot\|_2$ norm. Then $|\mathbf{a}^\top \mathbf{b} - \mathbf{a}'^\top \mathbf{b}'| \leq \|\mathbf{a} - \mathbf{a}'\|_2 + \|\mathbf{b} - \mathbf{b}'\|_2$.*

*Theorem* C.8 (Posterior log-odds perturbation for D²O+ADAPT). *Assume bounded features ($\|\mathbf{f}\|_2 \leq 1$) and well-conditioned covariance ($\|\boldsymbol{\Sigma}^{-1}\|_{\text{op}} \leq M_\Sigma$). Let $e = e(\mathbf{x})$ and let $\widehat{\mathbf{b}}_e$ satisfy Theorem C.4. Then for any pair $(y, k)$, with probability at*

*least $1 - \delta$, the perturbation $|\Delta_{k\to y}(\mathbf{x}) - \Delta^\star_{k\to y}(\mathbf{x})|$ is bounded by*

$$
\underbrace{2\gamma_{\mathrm{env}} \left\| \widehat{\mathbf{b}}_e - \mathbf{b}_e \right\|_\infty}_{\textit{prior debiasing error}}
$$

$$
+ \underbrace{\frac{1}{2s} \sum_{c \in \{y,k\}} \left( 4M_\Sigma R_c \left\| \mathbf{f} - \mathbf{f}^\star \right\|_2 + 4M_\Sigma R_c \left\| \boldsymbol{\mu}_c - \boldsymbol{\mu}_c^\star \right\|_2 + 4M_\Sigma^2 R_c^2 \left\| \boldsymbol{\Sigma} - \boldsymbol{\Sigma}^\star \right\|_{\mathrm{op}} \right)}_{\textit{Gaussian posterior error (using raw features } \mathbf{f}\textit{)}} \tag{27}
$$

$$
+ \underbrace{2\beta \left( \frac{2}{\tau_w} |\mathcal{B}| \left\| \mathbf{f}_{\mathrm{cnt}} - \mathbf{f}_{\mathrm{cnt}}^\star \right\|_2 + \frac{2}{\tau_w} \sum_{j \in \mathcal{B}} \left\| \mathbf{f}_{\mathrm{cnt},j} - \mathbf{f}_{\mathrm{cnt},j}^\star \right\|_2 + \sum_{j \in \mathcal{B}} \left\| \hat{\mathbf{p}}_j - \hat{\mathbf{p}}_j^\star \right\|_1 \right)}_{\textit{bank evidence error (using retrieval-oriented content features } \mathbf{f}_{\mathrm{cnt}}\textit{)}},
$$

*Here $\tau_w$ is the retrieval temperature and $R_c$ is a uniform residual bound for class $c$. The pseudo-label error satisfies, for each $j \in \mathcal{B}$,*

$$
\left\| \hat{\mathbf{p}}_j - \hat{\mathbf{p}}_j^\star \right\|_1 \le 2\gamma_{\mathrm{env}} \left\| \widehat{\mathbf{b}}_{e(\mathbf{x}_j)} - \mathbf{b}_{e(\mathbf{x}_j)} \right\|_\infty.
$$

*Corollary C.9 (Margin-based label invariance). Let $m(\mathbf{x}) := \min_{k \ne y} \Delta^\star_{k\to y}(\mathbf{x})$ be the oracle margin. If the perturbation bound in Theorem C.8 is at most $m(\mathbf{x})/2$, then $D^2O$+ADAPT predicts the same label as the oracle.*

# D. Proofs

## D.1. Proof of Lemma C.5 (Softmax Stability)

Let $\mathbf{w} = \mathcal{S}(\mathbf{a}/\tau)$ and $\mathbf{w}' = \mathcal{S}(\mathbf{a}'/\tau)$. The Jacobian is $J_{ij} = \frac{\partial w_i}{\partial a_j} = \frac{1}{\tau} w_i(\delta_{ij} - w_j)$. For any $i$, $\sum_j |J_{ij}| = \frac{1}{\tau} \sum_j w_i |\delta_{ij} - w_j| = \frac{1}{\tau}\left(w_i(1-w_i) + w_i \sum_{j \ne i} w_j\right) = \frac{2}{\tau} w_i(1-w_i) \le \frac{2}{\tau} w_i$. Thus,

$$
\left\| \mathbf{w} - \mathbf{w}' \right\|_1 \le \sum_i \sum_j \left| \frac{\partial w_i}{\partial a_j} \right| \left\| \mathbf{a} - \mathbf{a}' \right\|_\infty \le \frac{2}{\tau} \left( \sum_i w_i \right) \left\| \mathbf{a} - \mathbf{a}' \right\|_\infty = \frac{2}{\tau} \left\| \mathbf{a} - \mathbf{a}' \right\|_\infty.
$$

$\square$

## D.2. Proof of Lemma C.7 (Inner-Product Perturbation)

This follows directly from $|\mathbf{a}^\top \mathbf{b} - \mathbf{a}'^\top \mathbf{b}'| \le \left\| \mathbf{a} - \mathbf{a}' \right\|_2 \left\| \mathbf{b} \right\|_2 + \left\| \mathbf{a}' \right\|_2 \left\| \mathbf{b} - \mathbf{b}' \right\|_2$ under the unit-norm assumption. $\square$

## D.3. Proof of Lemma C.6 (Quadratic Discriminant Perturbation)

Decompose the difference as

$$
\mathbf{r}^\top \boldsymbol{\Sigma}^{-1} \mathbf{r} - \mathbf{r}'^\top \boldsymbol{\Sigma}'^{-1} \mathbf{r}' = \mathbf{r}^\top \boldsymbol{\Sigma}^{-1}(\mathbf{r} - \mathbf{r}') + (\mathbf{r} - \mathbf{r}')^\top \boldsymbol{\Sigma}^{-1} \mathbf{r}' + \mathbf{r}'^\top (\boldsymbol{\Sigma}^{-1} - \boldsymbol{\Sigma}'^{-1}) \mathbf{r}'.
$$

Using $\left\| \boldsymbol{\Sigma}^{-1} \right\|_2 \le M_\Sigma$ and $\left\| \mathbf{r} \right\| \le 2R$, the first two terms are bounded by $4M_\Sigma R \left\| \mathbf{r} - \mathbf{r}' \right\|_2$. The third term is bounded by $\left\| \mathbf{r}' \right\|^2 \left\| \boldsymbol{\Sigma}^{-1} \right\| \left\| \boldsymbol{\Sigma}'^{-1} \right\| \left\| \boldsymbol{\Sigma}' - \boldsymbol{\Sigma} \right\| \le 4R^2 M_\Sigma^2 \left\| \boldsymbol{\Sigma} - \boldsymbol{\Sigma}' \right\|_{\mathrm{op}}$. Substituting $\left\| \mathbf{r} - \mathbf{r}' \right\|_2 \le \left\| \mathbf{x} - \mathbf{x}' \right\|_2 + \left\| \boldsymbol{\mu} - \boldsymbol{\mu}' \right\|_2$ yields the result. $\square$

## D.4. Proof of Theorem C.3 (Subspace Recovery)

Using standard matrix concentration inequalities, the sub-Gaussian assumption on $\boldsymbol{\delta}$ gives a covariance concentration bound of order $\sqrt{1/N_\delta}$. Davis–Kahan then converts this operator-norm perturbation into a subspace-angle bound. Specifically, with probability $1 - \delta$,

$$
\| \sin\Theta(\widehat{\mathbf{V}}, \mathbf{V}_\star) \|_2 \le \frac{c\kappa^2}{\Delta} \sqrt{\frac{d + \log(1/\delta)}{N_\delta}}.
$$

This matches Eq. (23) up to the higher-order correction term written explicitly in the main statement. $\square$

## D.5. Proof of Theorem C.4 (Bias Estimation)

Apply Azuma–Hoeffding inequality to the martingale difference sequence induced by the EMA updates. The estimation error decomposes into the initialization bias decay term $(1 - \rho)^n$ and the accumulated noise term. The latter is bounded by $B\sqrt{2\log(2C/\delta)\sum \alpha_t^2}$. With EMA weights $\alpha_t = \rho(1 - \rho)^{n-t}$, the sum satisfies $\sum_t \alpha_t^2 \approx \frac{\rho}{2-\rho}$, which yields the effective sample size $n_{\mathrm{eff}}$. Routing errors contribute an additional bias term proportional to $p_{\mathrm{rt}}$. $\qquad\square$

## D.6. Proof of Theorem C.8 (Log-Odds Perturbation)

We bound the three terms in the log-odds decomposition separately.

**1. Prior term.** As derived in the main text, $|\Delta_{\mathrm{prior}} - \Delta_{\mathrm{prior}}^\star| \leq 2\gamma_{\mathrm{env}} \left\|\widehat{\mathbf{b}}_e - \mathbf{b}_e\right\|_\infty$.

**2. Gaussian term.** The Gaussian score uses **raw features f**, consistent with the design of D²O+ADAPT. For each class $c$, apply Lemma C.6 with the residual bound $R_c$. This gives

$$|\Delta_{\mathrm{GDA}} - \Delta_{\mathrm{GDA}}^\star| \leq \frac{1}{2s} \sum_{c\in\{y,k\}} \left(4M_\Sigma R_c \|\mathbf{f} - \mathbf{f}^\star\|_2 + 4M_\Sigma R_c \|\boldsymbol{\mu}_c - \boldsymbol{\mu}_c^\star\|_2 + 4M_\Sigma^2 R_c^2 \|\boldsymbol{\Sigma} - \boldsymbol{\Sigma}^\star\|_{\mathrm{op}}\right).$$

If raw feature extraction is deterministic, then $\mathbf{f} = \mathbf{f}^\star$, and the first term vanishes.

**3. Bank term.** The bank term is $\beta(g_y - g_k)$. We bound $|g_c - g_c^\star|$ where $g_c = \sum_{j\in\mathcal{B}} w_j \hat{p}_{j,c}$:

$$|g_c - g_c^\star| \leq \sum_{j\in\mathcal{B}} \left|w_j \hat{p}_{j,c} - w_j^\star \hat{p}_{j,c}^\star\right| \leq \sum_{j\in\mathcal{B}} \left(|w_j - w_j^\star| + |\hat{p}_{j,c} - \hat{p}_{j,c}^\star|\right).$$

**(i) Weight perturbation** $|w_j - w_j^\star|$. The weights $w_j = \mathcal{S}(\mathbf{sim}/\tau_w)_j$ are computed using the **retrieval-oriented content features $\mathbf{f}_{\mathrm{cnt}}$**. Let $\mathbf{a}$ be the vector of similarities, where $a_j = \mathbf{f}_{\mathrm{cnt}}^\top \mathbf{f}_{\mathrm{cnt},j}$. By Lemma C.7, $|a_j - a_j^\star| \leq \|\mathbf{f}_{\mathrm{cnt}} - \mathbf{f}_{\mathrm{cnt}}^\star\|_2 + \left\|\mathbf{f}_{\mathrm{cnt},j} - \mathbf{f}_{\mathrm{cnt},j}^\star\right\|_2$. By Lemma C.5,

$$\sum_{j\in\mathcal{B}} |w_j - w_j^\star| = \|\mathbf{w} - \mathbf{w}^\star\|_1 \leq \frac{2}{\tau_w} \max_j |a_j - a_j^\star|.$$

Using $\max_j u_j \leq \sum_j u_j$ for nonnegative $u_j$, we loosen the maximum over bank samples by a summation, which gives the bank evidence term in Theorem C.8.

**(ii) Label perturbation** $|\hat{p}_{j,c} - \hat{p}_{j,c}^\star|$. Using Lemma C.5 on the pseudo-labels $\hat{\mathbf{p}}_j = \mathcal{S}(\mathbf{s}_{\mathrm{deb}}(\mathbf{x}_j))$,

$$\sum_{j\in\mathcal{B}} |\hat{p}_{j,c} - \hat{p}_{j,c}^\star| \leq \sum_{j\in\mathcal{B}} \left\|\hat{\mathbf{p}}_j - \hat{\mathbf{p}}_j^\star\right\|_1 \leq \sum_{j\in\mathcal{B}} 2\gamma_{\mathrm{env}} \left\|\widehat{\mathbf{b}}_{e(\mathbf{x}_j)} - \mathbf{b}_{e(\mathbf{x}_j)}\right\|_\infty.$$

Combining (i) and (ii) yields the stated bank term bound. $\qquad\square$

# E. Implementation Details

In this section, we provide the concrete configurations, hyperparameters, and algorithmic details used to reproduce D²O.

## E.1. Finite-Difference Augmentation Specification

To estimate the nuisance-sensitive subspace via Eq. (3), we use a set of paired weak augmentations $\mathcal{T}$. The key idea is that these perturbations are mild enough to largely preserve semantic identity while inducing measurable variation along style–nuisance directions. In implementation, the finite-difference response is computed as

$$\boldsymbol{\delta} = \frac{\mathbf{f}(T_{+\epsilon}(\mathbf{x})) - \mathbf{f}(T_{-\epsilon}(\mathbf{x}))}{2\epsilon}.$$

We employ both photometric and geometric perturbations. For **ImageNet** variants, the perturbation magnitudes ($\epsilon$) are set as follows:

- **Photometric:** Brightness ($\epsilon = 0.06$), Contrast ($\epsilon = 0.08$), Saturation ($\epsilon = 0.08$).

- **Geometric:** Rotation ($\epsilon = 2.0°$ for standard, $3.0°$ for OOD), Scale ($\epsilon = 0.04$).

For **PUG-ImageNet**, we slightly adjust the perturbation strengths to better match the synthetic variation regime: Brightness (0.04), Contrast (0.06), Saturation (0.06), Rotation ($5.0°$), Scale (0.06), and vertical Translation (0.05 image height). Here, $\epsilon$ denotes the normalized perturbation magnitude (e.g., $x' = x \cdot (1 \pm \epsilon)$ for brightness).

**Computational accounting.** The warm-up cost is bounded by

$$N_{\text{warmup}} = N_S \times (2|\mathcal{T}| + 1)$$

forward passes. In our implementation, $|\mathcal{T}| = 5$ for ImageNet variants and $|\mathcal{T}| = 6$ for PUG-ImageNet. With $N_S = 128$, this corresponds to roughly 1400–1700 image forward passes, which is small relative to the full evaluation stream.

### E.2. Online Environment Centroid Maintenance

The routing clusters are maintained online using a running average update. For a routed cluster $k^*(\mathbf{x})$ with current centroid $\mathbf{c}_{k^*(\mathbf{x})}^{(t-1)}$ and count $n_{k^*(\mathbf{x})}$, after observing the style-aware routing coordinate $\mathbf{z}_{\text{sty}}(\mathbf{x})$, we update the centroid as

$$\mathbf{c}_{k^*(\mathbf{x})}^{(t)} = \frac{n_{k^*(\mathbf{x})}\, \mathbf{c}_{k^*(\mathbf{x})}^{(t-1)} + \mathbf{z}_{\text{sty}}(\mathbf{x})}{n_{k^*(\mathbf{x})} + 1}. \tag{28}$$

This update allows the centroids to stabilize progressively as more routing statistics are accumulated from the test stream. New clusters are initialized dynamically from the first sample assigned to them until the maximum routing capacity $K_{\text{env}} = 16$ is reached.

### E.3. Feature Choice for ImageNet-C Corruption Robustness

For the ImageNet-C corruption robustness experiments, we use the raw CLIP image feature $\mathbf{f}$ as the retrieval feature rather than the projected content feature $\mathbf{f}_{\text{cnt}}$. This is a conservative implementation choice because ImageNet-C mainly introduces low-level corruptions, where projection-based suppression may remove useful information together with corruption-sensitive directions. Accordingly, the retrieval-purity scores for ImageNet-C in Table 5 coincide across feature choices because they are evaluated under the same raw-feature retrieval setting.

## F. Additional Experimental Results

### F.1. Experiments with CLIP-RN50 Backbone

In this section, we further evaluate D²O+ADAPT using a CLIP-RN50 backbone under both **online** and **transductive** protocols. Across the ImageNet OOD suite (ImageNet-A/V/R/S), the method achieves the strongest average among the compared approaches in both settings, indicating that the benefits of D²O are not limited to a single backbone (Table 13). On corruption robustness (ImageNet-C), we again observe consistent gains in both online and transductive settings, although these gains remain more incremental than those on PUG-like shifts (Table 14). For the factorized PUG shifts, D²O+ADAPT achieves the clearest improvements across most factors, further supporting its usefulness under style/environment-dominant variation (Table 15). Finally, on fine-grained categorization benchmarks spanning diverse domains (e.g., Aircraft, DTD, EuroSAT, Food101, Pets, SUN397, and UCF101), the method remains competitive and attains the best average in both settings (Table 16). Overall, these RN50-based results support that the benefits of D²O are transferable across backbones and evaluation protocols.

### F.2. Experiments with Different Random Seeds

To verify the stability and reproducibility of D²O+ADAPT, we conduct experiments with three different random seeds (i.e., $\{0, 1, 2\}$) under both Online and Transductive settings. The evaluation covers ImageNet variants for OOD generalization, ImageNet-C for corruption robustness, ten fine-grained datasets, and PUG-ImageNet for nuisance-factor analysis.

**Online Setting.** As shown in Tables 17, 18, 19, and 20, D$^2$O+ADAPT remains highly stable across seeds in the online setting. For example, on ImageNet variants, the average OOD accuracy varies only slightly between 65.86% and 65.97%. A similarly small variance is observed on ImageNet-C and the fine-grained suite. These results indicate that the method is not sensitive to random initialization in the online regime.

**Transductive Setting.** We further evaluate seed sensitivity in the transductive setting, where the model adapts with access to the full unlabeled test set. Results in Tables 21, 22, 23, and 24 confirm that D$^2$O+ADAPT remains similarly stable in this setting. For instance, on the fine-grained suite, the average accuracy stays tightly clustered around 72.6%. Overall, these results support that the behavior of D$^2$O+ADAPT is reproducible across seeds in both evaluation protocols.

### F.3. Ablation on Warm-up Sampling Bias under Long-Tailed Streams

We investigate whether the warm-up set used for nuisance-sensitive subspace estimation is sensitive to sampling bias, particularly when the test stream exhibits a long-tailed class distribution. We consider three strategies: **Default**, which uses a warm-up prefix of $N_S$=128 test instances (effectively random sampling due to pre-shuffling); **Tail_only**, which samples the warm-up set exclusively from tail classes (defined by low test-time class frequency); and **Tail_mix**, which constructs a mixed warm-up set containing both tail and non-tail classes with a fixed mixing ratio.

As illustrated in Figure 6, performance remains highly stable across multiple benchmarks (ImageNet-A, ImageNet-V2, and PUG variants). Notably, even under the extreme *Tail_only* setting, the accuracy deviation is minimal relative to the *Default* setting. This suggests that the estimated nuisance-sensitive subspace captures global shift patterns rather than overfitting to the class composition of the warm-up buffer.

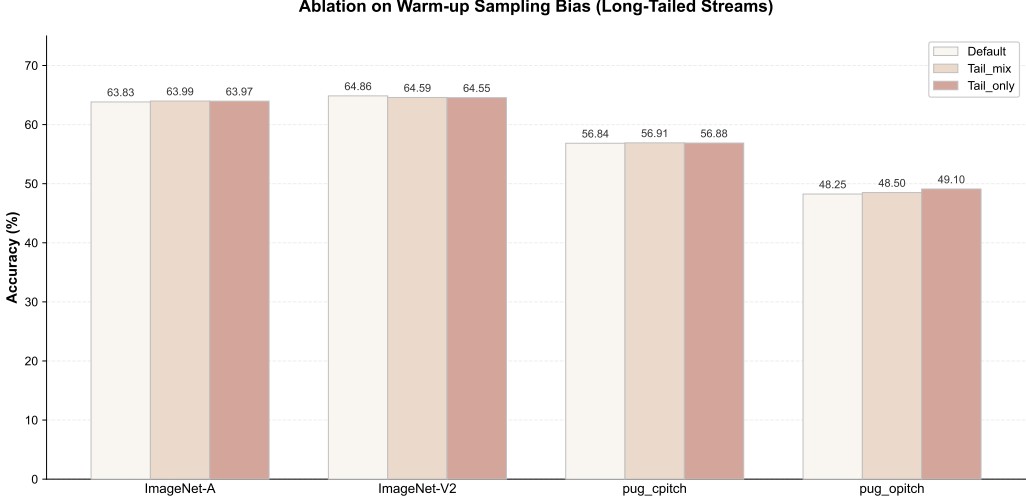

*Figure 6.* **Ablation on warm-up sampling bias under long-tailed streams.** We compare three warm-up sampling strategies: *Default* (random sampling), *Tail_mix* (mixed sampling), and *Tail_only* (sampling exclusively from tail classes). The consistent accuracy across ImageNet-A, ImageNet-V2, and PUG-ImageNet variants suggests that nuisance-sensitive subspace estimation is robust to class distribution bias in the warm-up buffer.

### F.4. Additional Qualitative Analysis of the Retrieval Mechanism

In the main paper, we report Top-5 retrieval purity (Pur@5) on PUG to illustrate the effect of retrieval-oriented content evidence. Here, we provide additional Pur@5 plots on other benchmarks, further corroborating that the retrieval-oriented content feature improves the semantic consistency of retrieved support samples under diverse shifts.

**Retrieval Purity on ImageNet Variants.** We further evaluate retrieval quality on four ImageNet variants with diverse distribution shifts. Pur@5 is defined as the fraction of Top-5 retrieved support samples sharing the same ground-truth label as the query. As shown in Figure 7, retrieval-oriented content features consistently increase purity on all variants, suggesting that retrieved evidence becomes more semantically aligned with the query even under challenging appearance changes such as sketch-like or artistic rendering shifts.

**Retrieval Purity on Fine-Grained Categorization.** To further assess the retrieval effect beyond PUG, we report Top-5 retrieval purity on ten fine-grained benchmarks. As shown in Figure 8, replacing raw retrieval keys with retrieval-oriented content features consistently increases purity across all datasets. This indicates that the retrieval process becomes less sensitive to nuisance appearance/environment factors and provides more semantically coherent support evidence for adaptation.

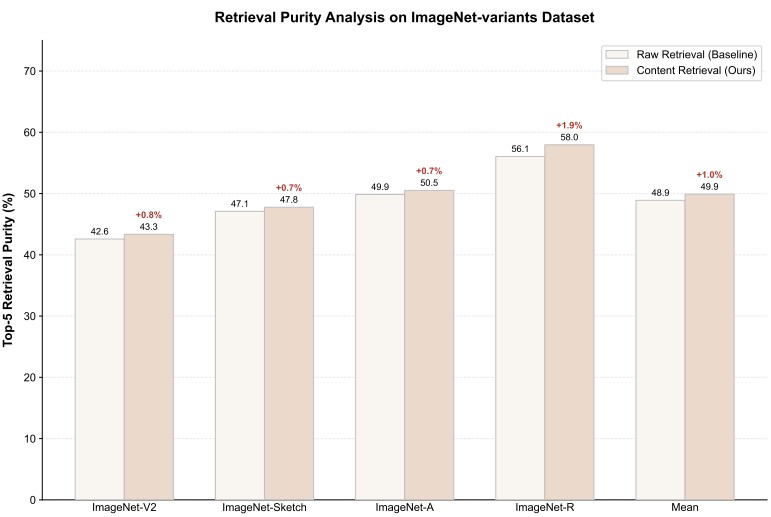

*Figure 7.* **Retrieval purity on ImageNet variants.** Retrieval-oriented content features consistently improve Top-5 retrieval purity (Pur@5) over raw feature retrieval across ImageNet-V2/Sketch/A/R, indicating reduced retrieval confounding under distribution shift.

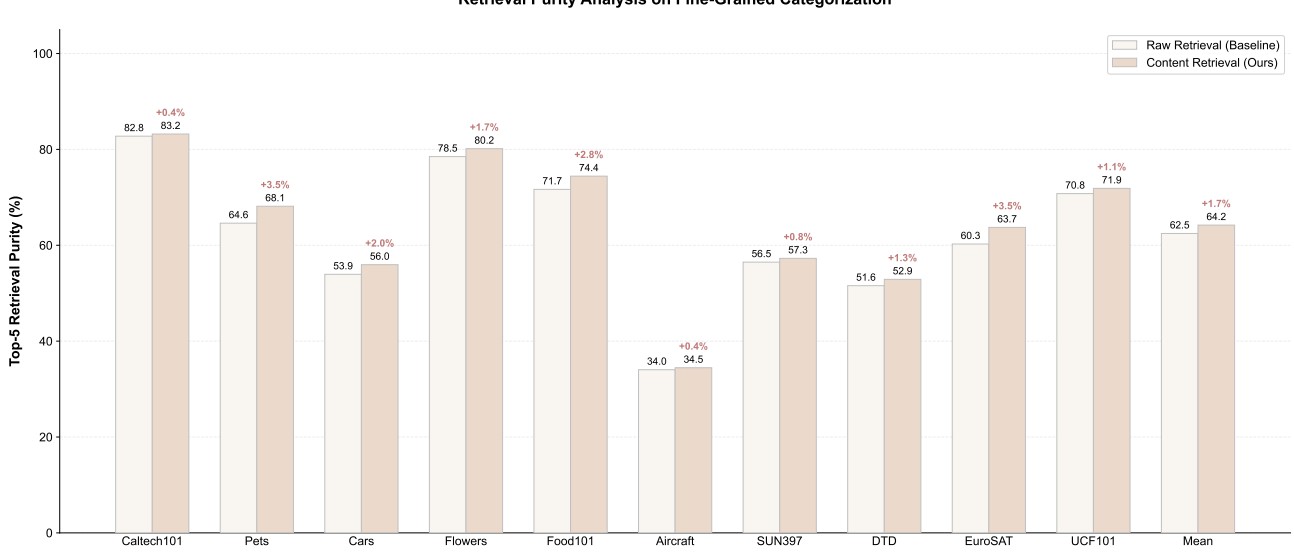

*Figure 8.* **Retrieval purity on fine-grained benchmarks.** Top-5 retrieval purity (Pur@5) of knowledge bank retrieval using raw features (baseline) versus retrieval-oriented content features (ours). Retrieval-oriented content features improve purity consistently across datasets, indicating reduced nuisance-driven retrieval confounding.

### F.5. Ablation on Nuisance-Sensitive Subspace Estimation: Finite-Difference Estimation vs. Simple PCA

A natural question regarding D²O is whether the finite-difference (FD) estimation of the nuisance-sensitive subspace $\widehat{\mathbf{V}}$ is necessary, or whether a simpler Principal Component Analysis (PCA) on the test batch would suffice. To investigate this, we implement a baseline that removes the top-$r$ principal components of the raw test features.

**Analysis: the trade-off between aggressiveness and safety.** Figure 9 reveals a clear trade-off.

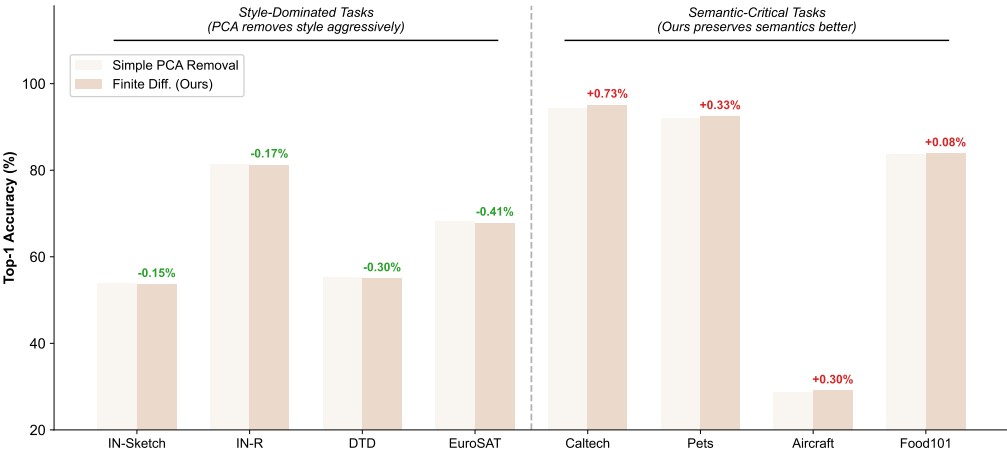

*Figure 9.* **Finite difference vs. simple PCA.** We compare Top-1 accuracy on style/nuisance-dominated benchmarks (left) and semantic-critical fine-grained benchmarks (right). While simple PCA can be competitive on texture-heavy datasets by aggressively removing dominant variance, it suffers from **semantic collapse** on fine-grained tasks (e.g., -0.73% on Caltech101), where high-variance directions may still contain class-discriminative information. By contrast, FD acts as a more reliable semantic safeguard across both regimes.

- **PCA is aggressive but blind.** On benchmarks dominated by strong global appearance shifts (e.g., ImageNet-Sketch, EuroSAT), PCA can marginally outperform FD (∼0.1–0.2%) because the dominant variance directions align closely with nuisance variation.

- **FD is more semantically conservative.** On semantic-critical benchmarks (e.g., Caltech101, OxfordPets, FGVC-Aircraft), blindly removing dominant variance degrades performance. For example, PCA drops accuracy by **0.73%** on Caltech101 and **0.33%** on OxfordPets relative to FD. This suggests that PCA can conflate intra-class semantic variation with nuisance directions.

**Conclusion.** Unlike PCA, FD estimates the subspace from paired augmentation differences, which are designed to suppress semantic identity while emphasizing nuisance-sensitive variation. This provides a more reliable safeguard against removing class-discriminative information and explains why FD generalizes better across heterogeneous tasks.

### F.6. Ablation Study: Feature Choice for GDA (Raw vs. Retrieval-oriented Content)

A core design choice of D²O+ADAPT is its **hybrid feature strategy**. We investigate the effect of feeding either raw features ($\mathbf{f}$) or retrieval-oriented content features ($\mathbf{f}_{\text{cnt}}$) into the GDA module. Table 9 reports the comparison across 14 benchmarks.

We observe a clear trade-off:

- **Natural and structure-dominated shifts.** On ImageNet-V2, ImageNet-Sketch, and ImageNet-R, raw features consistently outperform retrieval-oriented content features. This suggests that some appearance statistics (e.g., sketch stroke intensity or texture cues) remain informative for Gaussian alignment rather than being purely harmful.

- **Fine-grained recognition.** On the 10-dataset fine-grained suite, raw features achieve a slightly higher average accuracy (71.12% vs. 70.95%). This indicates that suppressing nuisance directions too aggressively can also remove high-frequency details that are useful for sub-class discrimination.

- **Adversarial and synthetic shifts.** On ImageNet-A and PUG-ImageNet, retrieval-oriented content features show a slight advantage (e.g., +0.37% on IN-A), suggesting that stronger suppression of nuisance variation can help under more adversarial or synthetic shifts.

**Decision.** Since the gains from retrieval-oriented content features on adversarial settings are relatively small, whereas raw features provide stronger stability on natural shifts and fine-grained tasks, we adopt **raw features** as the default input to the

GDA branch. This hybrid choice is consistent with the overall design of D²O: nuisance-sensitive directions are suppressed where they are harmful for retrieval, but not uniformly removed from every inference component.

*Table 9.* **Feature choice for GDA: raw (f) vs. retrieval-oriented content ($f_{cnt}$).** Top-1 accuracy (%) comparison. Raw features achieve stronger performance on natural shifts (V2, Sketch, R) and fine-grained tasks, yielding the best overall trade-off for the Gaussian branch.

| GDA Input | IN-V2 | IN-Sketch | IN-R | IN-A | PUG-Avg[†] | Fine-Grained Avg[‡] | Overall Avg |
|---|---|---|---|---|---|---|---|
| Content ($f_{cnt}$) | 64.70 | 53.65 | 81.14 | **64.20** | **58.35** | 70.95 | 65.50 |
| Raw (f) (ours) | **64.86** | **53.69** | **81.29** | 63.83 | 58.28 | **71.12** | **65.51** |

[†] *PUG-Avg includes all 10 PUG-ImageNet nuisance factors.*
[‡] *Fine-Grained Avg includes 10 datasets: Caltech, Pets, Cars, Flowers, Food, Aircraft, SUN397, DTD, EuroSAT, UCF101.*

### F.7. Statistical Visualization of Centered Logit Bias under Environment Clusters

To empirically validate the corrected prior formulation $s_{deb}(\mathbf{x}) = s_{clip}(\mathbf{x}) - \gamma_{env} \mathbf{b}^{(k^*(\mathbf{x}))}$ (Eq. 12), we analyze online inference dynamics on **PUG-CameraRoll**, where rotational distortion serves as a challenging testbed for systematic environment-induced hallucination.

As illustrated in Figure 10, the effectiveness of this additive correction is supported by two observations. First, the norm of the bias template $\|\mathbf{b}^{(k)}\|_2$ (left panel) exhibits a clear rise-and-plateau pattern, quickly increasing to approximately 23.0 and then stabilizing. This suggests that the subtracted term captures a persistent environment-dependent relative logit distortion rather than random noise. The dominant aligned classes, such as **Class 65 ("Joystick")** and **Class 133 ("Torch")**, further indicate that $\mathbf{b}^{(k)}$ captures systematic rotational hallucinations.

Second, the entropy dynamics (right panel) show why this correction matters. Although the raw CLIP logits remain overconfident under severe rotational distortion, subtracting the estimated centered-logit bias raises the mean entropy from 1.28 to 1.52. This increase of **+0.23** suggests that the debiasing step acts as a safety mechanism, pulling overly confident incorrect predictions back toward a more uncertain state and thereby reducing the risk of error amplification.

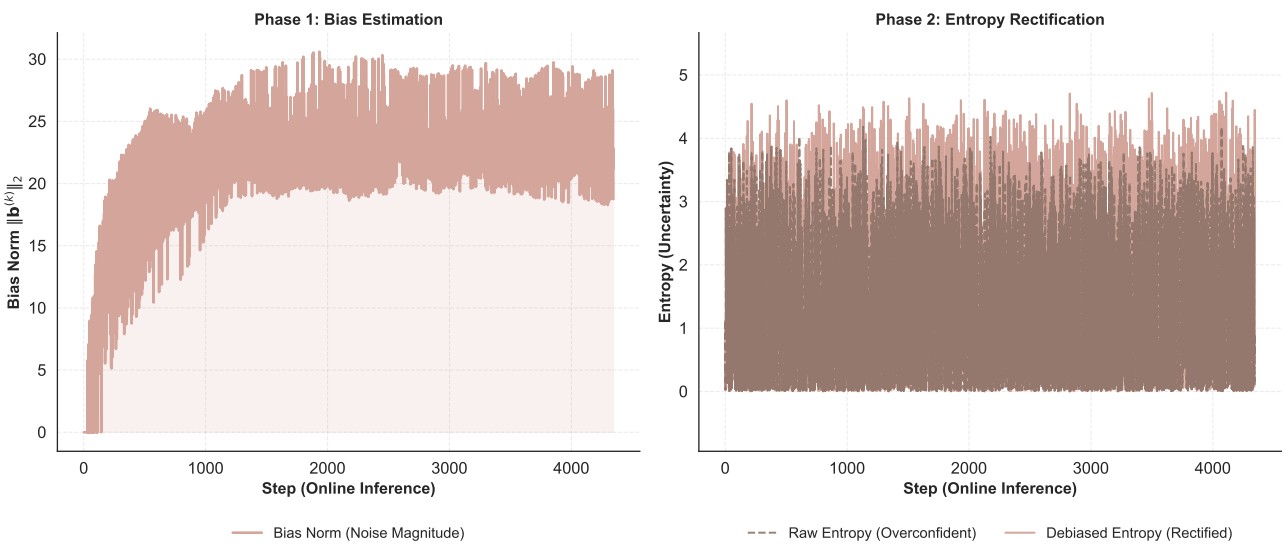

*Figure 10.* **Phenomenological evidence of centered-logit bias dynamics on PUG-CameraRoll. (Left)** The norm of the bias template $\|\mathbf{b}^{(k)}\|_2$ rapidly converges to $\sim 23.0$, indicating a stable environment-dependent distortion pattern. **(Right)** D²O raises the mean prediction entropy from 1.28 to 1.52 (**+0.23**) under rotational distortion, reducing raw overconfidence.

### F.8. Empirical Verification of Theoretical Assumptions

In this section, we provide empirical evidence for the two main assumptions used in Appendix C: the low-rank nature of environment-induced nuisance variation and the semantic safety of the estimated nuisance-sensitive subspace.

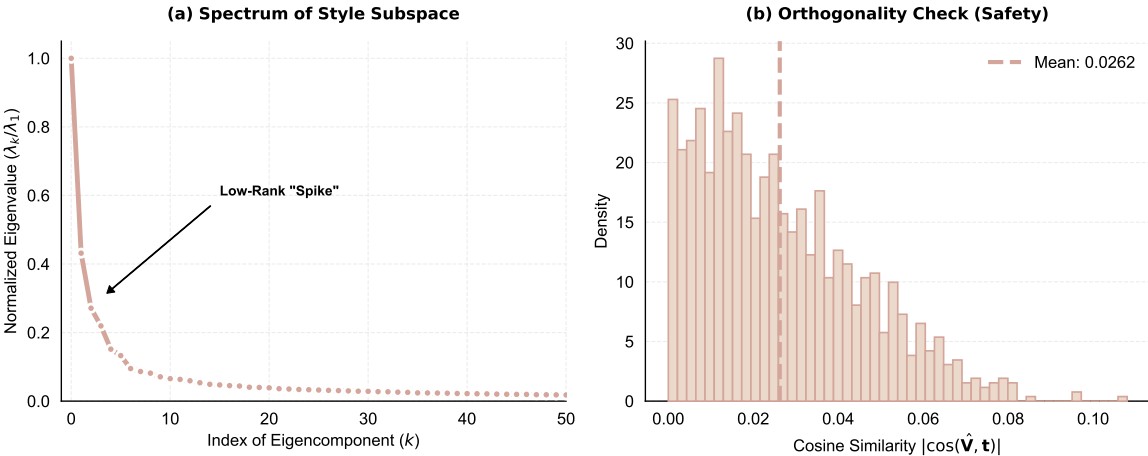

*Figure 11.* **Empirical verification of the theoretical assumptions on PUG-ImageNet. (a) Spiked covariance structure.** The eigenvalue spectrum of the estimated nuisance-sensitive subspace exhibits a sharp "L-shaped" decay. The marked "low-rank spike" ($\lambda_r$) separates dominant nuisance variation from the trailing noise floor ($\sigma^2$), supporting the low-rank approximation. **(b) Orthogonality / safety.** The histogram of cosine similarities between the estimated basis $\hat{\mathbf{V}}$ and the semantic prototypes $\mathbf{W}$ is concentrated near zero (mean $\approx 0.026$). This suggests that the estimated nuisance-sensitive directions are nearly orthogonal to the semantic decision boundary.

**Spectrum analysis.** As shown in Figure 11(a), the eigenvalues decay rapidly, indicating that environment-induced variation (e.g., Camera Pitch) is concentrated in a low-dimensional subspace rather than being diffuse. This supports the *spiked response covariance* assumption used in the theory.

**Orthogonality check.** Figure 11(b) shows that the estimated nuisance-sensitive directions are nearly orthogonal to the semantic prototypes. This suggests that projecting along the estimated subspace does not strongly interfere with the semantic decision boundary, which is consistent with the safety requirement in the *Confounded Feature Model*.

### F.9. Detailed Scope Evidence

In this section, we provide detailed results corresponding to the scope evidence summarized in Section 4.4. Tables 10, 11, and 12 expand the host family scope results on PUG, ImageNet-4, and Fine-Grained-10. For each host pair, the better value between the base host and its D²O-enhanced variant is marked in bold.

*Table 10.* **Detailed scope extension results on PUG.** We report factor-wise accuracies before and after inserting $D^2O$ into additional host families. For each host pair, the better value is marked in bold.

| Method | Cam-Yaw | Cam-Pitch | Cam-Roll | Pose-Yaw | Pose-Pitch | Pose-Roll | Scale | Texture | Lighting | Worlds | Avg. |
|---|---|---|---|---|---|---|---|---|---|---|---|
| CoOp | 43.14 | 35.13 | 32.14 | 42.13 | 25.76 | 24.38 | 42.75 | 28.85 | 4.24 | 32.29 | 31.08 |
| CoOp + $D^2O$ | **48.55** | **36.60** | **36.46** | **47.26** | **30.23** | **28.25** | **47.65** | **32.47** | **26.16** | **35.16** | **36.88** |
| MTA | 41.90 | **33.33** | 29.74 | 39.76 | **24.56** | **23.41** | 40.52 | 28.41 | 6.59 | **32.22** | 30.04 |
| MTA + $D^2O$ | **42.52** | 33.03 | **30.55** | **40.54** | 24.22 | 23.25 | **41.32** | **29.13** | **7.73** | 31.50 | **30.38** |
| ZERO | 45.44 | **36.05** | 33.63 | 43.72 | 27.60 | 26.61 | 44.50 | 31.69 | 15.17 | **39.49** | 34.39 |
| ZERO + $D^2O$ | **46.41** | 35.80 | **34.28** | **44.11** | **27.97** | **26.70** | **45.53** | **32.11** | **16.89** | 39.46 | **34.93** |
| TPT | 38.97 | 30.57 | 26.24 | 37.32 | 22.81 | 21.09 | 37.15 | 26.78 | 6.24 | 26.98 | 27.42 |
| TPT + $D^2O$ | **44.87** | **34.25** | **33.98** | **43.44** | **27.53** | **25.00** | **44.80** | **32.23** | **16.85** | **33.63** | **33.66** |

*Table 11.* **Detailed scope extension results on ImageNet-4.** We report accuracies on ImageNet-V2, ImageNet-Sketch, ImageNet-A, and ImageNet-R. For each host pair, the better value is marked in bold.

| Method | ImageNet-V2 | ImageNet-Sketch | ImageNet-A | ImageNet-R | Avg. |
|---|---|---|---|---|---|
| CoOp | 64.20 | 46.30 | 48.90 | 74.60 | 58.50 |
| CoOp + D²O | **64.42** | **49.77** | **49.48** | **76.62** | **60.07** |
| MTA | **63.67** | 48.51 | 57.23 | 76.98 | 61.60 |
| MTA + D²O | 63.57 | **50.76** | **58.23** | **77.88** | **62.61** |
| ZERO | **65.41** | 50.55 | 62.83 | 80.63 | 64.86 |
| ZERO + D²O | 65.14 | **50.89** | **63.03** | **80.87** | **64.98** |
| TPT | 63.45 | 47.94 | 54.77 | **77.06** | 60.81 |
| TPT + D²O | **65.02** | **50.38** | **62.53** | 76.48 | **63.61** |

*Table 12.* **Detailed scope extension results on Fine-Grained-10.** We report accuracies before and after inserting D²O into additional host families. For each host pair, the better value is marked in bold.

| Method | Aircraft | Caltech | Cars | DTD | EuroSAT | Flower | Food101 | Pets | SUN397 | UCF101 | Avg. |
|---|---|---|---|---|---|---|---|---|---|---|---|
| CoOp | 17.40 | **92.20** | 63.10 | 40.30 | 44.30 | 60.90 | 84.60 | 87.60 | 61.20 | 66.10 | 61.77 |
| CoOp + D²O | **17.88** | 92.13 | **66.52** | **43.09** | **54.96** | **63.87** | **85.34** | **88.91** | **64.83** | **69.60** | **64.71** |
| MTA | 27.69 | **95.21** | 65.60 | 52.90 | 55.43 | 73.85 | **83.99** | 90.62 | 67.97 | 67.96 | 68.12 |
| MTA + D²O | **28.32** | 94.77 | **68.54** | **54.67** | **67.69** | **76.05** | 83.87 | **92.15** | **70.40** | **71.66** | **70.81** |
| ZERO | 27.66 | **95.21** | 65.63 | 52.90 | 55.51 | 73.85 | **84.00** | 90.57 | 67.95 | 67.96 | 68.12 |
| ZERO + D²O | **28.41** | 94.77 | **68.51** | **54.73** | **67.83** | **76.05** | 83.91 | **92.12** | **70.40** | **71.61** | **70.83** |
| TPT | **24.78** | 94.16 | 66.87 | 47.75 | **42.44** | 68.98 | **84.67** | 87.79 | 65.50 | 68.04 | 65.10 |
| TPT + D²O | 24.24 | **94.52** | **69.84** | **48.76** | 40.42 | **70.36** | 82.42 | **88.47** | **67.78** | **70.24** | **65.71** |

*Table 13.* Online and transductive evaluation on ImageNet-A/V/R/S with a CLIP-RN50 backbone. "BP-free" indicates whether the method avoids backpropagation at test time.

| Setting | Method | BP-free | ImageNet-A | ImageNet-V | ImageNet-R | ImageNet-S | Avg. |
|---|---|---|---|---|---|---|---|
| | CLIP | – | 21.83 | 51.41 | 56.15 | 33.37 | 40.69 |
| | TPT | ✗ | 26.67 | 54.70 | 59.11 | 35.09 | 43.89 |
| | DiffTPT | ✗ | 31.06 | 55.80 | 58.80 | 37.10 | 45.69 |
| | C-TPT | ✗ | 23.40 | 54.70 | 58.00 | 35.10 | 42.80 |
| | DMN | ✗ | 28.57 | 56.12 | 61.44 | 39.84 | 46.49 |
| | TPS | ✗ | 29.80 | **60.04** | 55.49 | 35.74 | 45.27 |
| | DPE | ✗ | 30.15 | 56.72 | **63.72** | 40.03 | 47.66 |
| Online | DynaPrompt | ✗ | 27.84 | 55.12 | 60.63 | 35.64 | 44.81 |
| | BCA | ✓ | 30.35 | 56.58 | 62.89 | 38.04 | 46.97 |
| | Dota | ✓ | 30.81 | 55.27 | 62.81 | 37.52 | 46.60 |
| | TDA | ✓ | 30.29 | 55.54 | 62.58 | 38.12 | 46.63 |
| | ADAPT | ✓ | 33.08 | 55.97 | 62.69 | 40.21 | 47.99 |
| | D²O+ADAPT(ours) | ✓ | **33.57** | 56.22 | 62.93 | **40.67** | **48.35** (+0.36) |
| | TransCLIP | ✓ | 21.93 | 51.54 | 35.15 | **52.79** | 40.35 |
| Trans. | ADAPT | ✓ | 33.72 | 56.57 | 63.11 | 41.19 | 48.65 |
| | D²O+ADAPT(ours) | ✓ | **34.34** | **57.48** | **63.63** | 41.87 | **49.33** (+0.68) |

*Table 14.* Online and transductive corruption robustness on ImageNet-C with a CLIP-RN50 backbone. Best results are in **bold** and second-best are underlined. "BP-free" indicates whether the method avoids backpropagation at test time.

| Setting | Method | BP-free | Defo. | Glas. | Moti. | Zoom | Snow | Fros. | Fog | Brig. | Cont. | Elas. | Pix. | JPEG | Gauss. | Shot | Impu. | Avg. |
|---|---|---|---|---|---|---|---|---|---|---|---|---|---|---|---|---|---|---|
| Online | CLIP | – | 9.54 | 3.40 | 7.46 | 12.62 | 12.29 | 15.72 | 22.08 | 41.69 | 6.24 | 4.67 | 11.01 | 14.24 | 2.43 | 3.07 | 2.52 | 11.27 |
| | TPT | ✗ | 8.02 | 2.74 | 5.34 | 10.97 | 10.59 | 12.92 | 16.17 | 35.67 | 4.45 | 3.73 | 11.56 | **16.68** | 1.43 | 1.94 | 1.42 | 9.58 |
| | DiffTPT | ✗ | 10.50 | 3.90 | 8.62 | 12.74 | 11.31 | 15.03 | 19.64 | 37.73 | 4.98 | 4.74 | 13.58 | 16.56 | 2.69 | 3.59 | 2.08 | 11.18 |
| | TDA | ✓ | 9.84 | 4.40 | 7.38 | 13.74 | 13.74 | 17.16 | 23.76 | 44.16 | 7.00 | 5.79 | 11.24 | 15.26 | 2.54 | 3.26 | 2.72 | 12.13 |
| | ADAPT | ✓ | 10.54 | 4.44 | 8.57 | 14.34 | 13.85 | 17.84 | 24.56 | 45.67 | 7.76 | 5.85 | 11.96 | 15.86 | 2.91 | 3.77 | 2.92 | 12.72 |
| | D²O+ADAPT(ours) | ✓ | **10.57** | **4.51** | **8.74** | **14.37** | 13.84 | **17.94** | **24.62** | **45.86** | **7.80** | **5.94** | 12.13 | 15.80 | **2.93** | 3.56 | **3.00** | **12.77** (+0.05) |
| Trans. | ZLaP | ✓ | 10.30 | 3.54 | 7.99 | 13.47 | 13.66 | 17.15 | 23.20 | 44.67 | 6.55 | 5.15 | 11.61 | 14.23 | 1.17 | 2.33 | 1.65 | 11.78 |
| | TransCLIP | ✓ | 9.38 | 4.17 | 7.14 | 12.42 | 11.87 | 15.04 | 23.93 | 44.33 | 6.45 | 6.11 | 11.03 | 14.85 | 2.87 | 3.14 | 3.13 | 11.72 |
| | ADAPT | ✓ | 12.26 | 6.11 | 10.92 | 17.02 | 16.67 | 20.62 | 27.77 | 47.31 | 9.20 | 8.56 | 14.61 | 18.15 | 4.05 | 4.80 | 4.03 | 14.81 |
| | D²O+ADAPT(ours) | ✓ | **12.83** | **6.76** | **11.63** | **18.13** | **17.96** | **22.04** | **29.58** | **48.36** | **9.82** | **10.37** | **15.47** | **18.83** | **4.46** | **5.37** | **4.58** | **15.75** (+0.94) |

*Table 15.* Online and transductive results on PUG datasets with a CLIP-RN50 backbone. Best results are in **bold** and second-best are underlined. "BP-free" indicates whether the method avoids backpropagation at test time.

| Setting | Method | BP-free | Camera (Yaw / Pitch / Roll) | Pose (Yaw / Pitch / Roll) | Scale | Texture | Lighting | Worlds | Avg. |
|---|---|---|---|---|---|---|---|---|---|
| Online | DMN-ZS | ✗ | 50.07/37.62/35.43 | 50.83/29.77/29.97 | 47.54 | 35.85 | 16.57 | 24.79 | 35.84 |
| | ADAPT | ✓ | 54.95/43.58/43.19 | 54.47/34.02/33.20 | 54.67 | 39.20 | 21.53 | 40.39 | 41.92 |
| | D²O+ADAPT(ours) | ✓ | **61.88/47.21/47.01** | **61.67/40.06/38.81** | **61.00** | **45.06** | **36.33** | **45.47** | **48.45** (+6.53) |
| Trans. | ADAPT | ✓ | 56.31/45.14/45.67 | 56.28/36.90/35.13 | 56.45 | 41.51 | 22.55 | 41.60 | 43.75 |
| | D²O+ADAPT(ours) | ✓ | **60.77/47.33/49.03** | **60.70/40.70/38.70** | **60.29** | **45.63** | **33.29** | **43.38** | **47.98** (+4.23) |

*Table 16.* Online and transductive results on fine-grained categorization benchmarks with a CLIP-RN50 backbone. Best results are in **bold** and second-best are underlined. "BP-free" indicates whether the method avoids backpropagation at test time.

| Setting | Method | BP-free | Aircraft | Caltech | Cars | DTD | EuroSAT | Flower | Food101 | Pets | Sun397 | UCF101 | Avg. |
|---|---|---|---|---|---|---|---|---|---|---|---|---|---|
| Online | CLIP | – | 15.66 | 85.88 | 55.70 | 40.37 | 23.69 | 61.75 | 73.97 | 83.57 | 58.80 | 58.84 | 55.82 |
| | TPT | ✗ | 17.58 | 87.02 | 58.46 | 40.84 | 28.33 | 62.69 | 74.88 | 84.49 | 61.46 | 60.82 | 57.66 |
| | DiffTPT | ✗ | 17.60 | 86.89 | **60.71** | 40.72 | 41.04 | 63.53 | **79.21** | 83.40 | 62.72 | 62.67 | 59.85 |
| | C-TPT | ✗ | 17.00 | 86.90 | 56.50 | 42.20 | 27.80 | 65.20 | 74.70 | 84.10 | 61.00 | 59.70 | 57.51 |
| | DIN | ✗ | **22.77** | 90.14 | 60.02 | 50.41 | 48.72 | 67.93 | 76.70 | 86.78 | 64.39 | **65.34** | 63.32 |
| | TPS | ✗ | 18.30 | 89.80 | 59.40 | 48.40 | 24.30 | 68.20 | 76.20 | 84.40 | 62.70 | 64.30 | 59.60 |
| | DPE | ✗ | 19.80 | **90.83** | 59.26 | 50.18 | 41.67 | 67.60 | 77.83 | 85.97 | 64.23 | 61.98 | 61.94 |
| | BCA | ✓ | 19.89 | 89.70 | 58.13 | 48.58 | 42.12 | 66.30 | 77.19 | 85.58 | 63.38 | 63.51 | 61.44 |
| | TDA | ✓ | 17.61 | 89.70 | 57.78 | 43.74 | 42.11 | 68.74 | 77.75 | 86.18 | 62.53 | 64.18 | 61.03 |
| | Dota | ✓ | 18.06 | 88.84 | 58.72 | 45.80 | 47.15 | 68.53 | 78.61 | **87.33** | 63.89 | 65.08 | 62.20 |
| | ADAPT | ✓ | 18.00 | 89.37 | 58.38 | 51.89 | 50.47 | 70.04 | 75.57 | 86.43 | 64.94 | 63.12 | 62.82 |
| | D²O+ADAPT (ours) | ✓ | 18.27 | 89.70 | 58.74 | **52.67** | **51.89** | **70.93** | 75.22 | 87.22 | **64.96** | 63.71 | **63.33** (+0.51) |
| Trans. | TransCLIP | ✓ | 16.60 | 88.60 | 57.90 | 47.80 | **59.60** | 72.20 | **78.00** | 89.30 | 64.20 | **68.80** | 64.30 |
| | StatA | ✓ | 16.00 | 87.30 | 58.20 | 48.50 | 50.50 | 67.70 | 77.90 | 87.70 | 64.30 | 67.50 | 62.56 |
| | ADAPT | ✓ | 19.53 | 90.99 | 61.46 | 55.73 | 46.26 | **74.14** | 76.97 | 87.95 | 66.66 | 66.75 | 64.64 |
| | D²O+ADAPT(ours) | ✓ | **19.77** | **92.09** | **62.02** | 54.96 | 47.38 | 73.57 | 76.88 | **88.42** | **66.69** | 67.51 | **64.93** (+0.29) |

*Table 17.* Performance on ImageNet variants across different random seeds in the **Online** setting. Best results are in **bold** and second-best are underlined.

| Seed | Method | ImageNet-V2 | ImageNet-Sketch | ImageNet-A | ImageNet-R | OOD Avg. |
|---|---|---|---|---|---|---|
| 0 | D²O+ADAPT | **64.86** | **53.69** | 63.83 | **81.29** | 65.92 |
| 1 | D²O+ADAPT | 64.67 | 53.60 | 64.39 | 81.22 | **65.97** |
| 2 | D²O+ADAPT | 64.12 | 53.51 | **64.52** | 81.27 | 65.86 |

*Table 18.* Robustness on ImageNet-C (Noise, Blur, Weather, Digital) across different random seeds in the **Online** setting. Best results are in **bold** and second-best are underlined.

| Seed | Method | Blur | | | | Weather | | | | Digital | | | | Noise | | | Avg. |
| | | Defo. | Glas. | Moti. | Zoom | Snow | Fros. | Fog | Brig. | Cont. | Elas. | Pix. | JPEG | Gauss. | Shot | Impu. | |
|---|---|---|---|---|---|---|---|---|---|---|---|---|---|---|---|---|---|
| 0 | D²O+ADAPT | **26.94** | 18.35 | 27.84 | 26.01 | 37.00 | 35.23 | 41.63 | 60.56 | 20.38 | 16.92 | 38.04 | 37.60 | **16.51** | **17.45** | **16.79** | 29.15 |
| 1 | D²O+ADAPT | 26.80 | **18.39** | **27.96** | 26.08 | 36.95 | 35.29 | 41.44 | 60.43 | 20.50 | 16.83 | 38.00 | 37.59 | 16.43 | 17.40 | 16.74 | 29.12 |
| 2 | D²O+ADAPT | 26.91 | 18.38 | 27.85 | **26.15** | **37.02** | 35.27 | **41.64** | **60.57** | 20.28 | **16.95** | **38.10** | 37.77 | 16.41 | 17.36 | 16.70 | **29.16** |

*Table 19.* Performance on Fine-grained datasets across different random seeds in the **Online** setting. Best results are in **bold** and second-best are underlined.

| Seed | Method | Caltech | Pets | Cars | Flowers | Food101 | Aircraft | SUN397 | DTD | EuroSAT | UCF101 | Avg. |
|---|---|---|---|---|---|---|---|---|---|---|---|---|
| 0 | D²O+ADAPT | **95.13** | 92.48 | 68.91 | **76.74** | **83.90** | 29.10 | **70.56** | **55.02** | 67.86 | **71.45** | **71.12** |
| 1 | D²O+ADAPT | 94.40 | 91.88 | **68.95** | 75.88 | 83.83 | **29.16** | 70.55 | 54.31 | 67.32 | 70.53 | 70.68 |
| 2 | D²O+ADAPT | 94.81 | **92.56** | 68.80 | 76.13 | 83.89 | **29.16** | 70.40 | 54.49 | **67.95** | 71.00 | 70.92 |

*Table 20.* Performance on PUG nuisance factors across different random seeds in the **Online** setting. Metrics for Camera and Pose are reported as (Yaw / Pitch / Roll). Best results are in **bold** and second-best are underlined.

| Seed | Method | Camera (Yaw / Pitch / Roll) | Pose (Yaw / Pitch / Roll) | Scale | Texture | Lighting | Worlds | Avg. |
|---|---|---|---|---|---|---|---|---|
| 0 | D²O+ADAPT | 71.52 / **56.84** / 58.29 | **70.99** / 48.25 / 47.03 | 69.71 | **56.78** | 43.88 | **59.55** | **58.28** |
| 1 | D²O+ADAPT | **71.64** / 56.22 / 58.03 | 70.83 / **48.37** / 47.40 | **69.84** | 55.77 | 43.21 | 59.39 | 58.07 |
| 2 | D²O+ADAPT | 70.61 / 56.47 / **58.36** | 70.83 / 48.30 / **47.47** | 69.41 | 55.94 | **44.73** | 58.87 | 58.10 |

*Table 21.* Performance on ImageNet variants across different random seeds in the **Transductive** setting. Best results are in **bold** and second-best are underlined.

| Seed | Method | ImageNet-V2 | ImageNet-Sketch | ImageNet-A | ImageNet-R | OOD Avg. |
|---|---|---|---|---|---|---|
| 0 | D²O+ADAPT | **66.18** | 54.24 | 64.67 | 81.01 | **66.53** |
| 1 | D²O+ADAPT | 65.66 | 54.04 | 64.41 | **81.12** | 66.31 |
| 2 | D²O+ADAPT | 65.96 | **54.41** | **64.68** | 81.05 | **66.53** |

*Table 22.* Robustness on ImageNet-C (Noise, Blur, Weather, Digital) across different random seeds in the **Transductive** setting. Best results are in **bold** and second-best are underlined.

| Seed | Method | Blur | | | | Weather | | | | Digital | | | | Noise | | | Avg. |
| | | Defo. | Glas. | Moti. | Zoom | Snow | Fros. | Fog | Brig. | Cont. | Elas. | Pix. | JPEG | Gauss. | Shot | Impu. | |
|---|---|---|---|---|---|---|---|---|---|---|---|---|---|---|---|---|---|
| 0 | D²O+ADAPT | 28.85 | **20.61** | 29.87 | 28.32 | 39.13 | **37.50** | 43.73 | **61.71** | **22.43** | **20.05** | 39.89 | **38.69** | 18.75 | **19.81** | 19.68 | **31.27** |
| 1 | D²O+ADAPT | **28.90** | 20.57 | 29.83 | **28.41** | **39.20** | 37.42 | 43.69 | 61.64 | 22.38 | 20.04 | 39.94 | 38.67 | 18.75 | 19.80 | 19.65 | 31.26 |
| 2 | D²O+ADAPT | 28.87 | 20.49 | **29.89** | 28.34 | 39.16 | 37.43 | **43.79** | 61.56 | 22.31 | 19.98 | **39.96** | 38.65 | **18.77** | 19.76 | **19.71** | 31.24 |

*Table 23.* Performance on Fine-grained datasets across different random seeds in the **Transductive** setting. Best results are in **bold** and second-best are underlined.

| Seed | Method | Caltech | Pets | Cars | Flowers | Food101 | Aircraft | SUN397 | DTD | EuroSAT | UCF101 | Avg. |
|---|---|---|---|---|---|---|---|---|---|---|---|---|
| 0 | D²O+ADAPT | 95.98 | 92.61 | **71.40** | 80.55 | 85.17 | 30.78 | 72.40 | 56.91 | **66.36** | **74.02** | **72.62** |
| 1 | D²O+ADAPT | **96.06** | **92.75** | 71.36 | 80.39 | **85.21** | 30.60 | **72.43** | 56.74 | 66.11 | 73.96 | 72.56 |
| 2 | D²O+ADAPT | 95.90 | 92.50 | 71.37 | **80.63** | 85.19 | **31.08** | 72.35 | **56.97** | 66.16 | 73.94 | 72.61 |

*Table 24.* Performance on PUG nuisance factors across different random seeds in the **Transductive** setting. Metrics for Camera and Pose are reported as (Yaw / Pitch / Roll). Best results are in **bold** and second-best are underlined.

| Seed | Method | Camera (Yaw / Pitch / Roll) | Pose (Yaw / Pitch / Roll) | Scale | Texture | Lighting | Worlds | Avg. |
|---|---|---|---|---|---|---|---|---|
| 0 | D²O+ADAPT | 70.65 / 55.87 / 57.82 | 69.89 / **48.57** / 45.99 | 68.64 | 54.84 | 39.90 | 56.33 | 56.85 |
| 1 | D²O+ADAPT | **71.22** / 56.72 / 58.75 | 70.26 / 48.37 / **46.64** | 68.53 | 54.73 | **40.33** | **56.41** | **57.20** |
| 2 | D²O+ADAPT | 70.88 / **56.74** / **59.23** | 70.44 / 47.86 / 46.50 | **68.90** | **55.20** | 39.60 | 56.20 | 57.15 |

